# Lipid-mediated gating of a miniature mechanosensitive MscS channel from *Trypanosoma cruzi*

Jingying Zhang[1,2,7], Aashish Bhatt[3,4,7], Grigory Maksaev ●[5,6,7], Yun Lyna Luo ●[3,4] ✉ & Peng Yuan ●[1,2] ✉

The mechanosensitive channel of small conductance (MscS) from *E. coli* (*Ec*MscS) has served as the prevailing model system for understanding mechanotransduction in ion channels. *Trypanosoma cruzi*, the protozoan parasite causing Chagas disease, encodes a miniature MscS ortholog (*Tc*MscS) critical for parasite development and infectivity. *Tc*MscS contains a minimal portion of the canonical *Ec*MscS fold yet maintains mechanosensitive channel activity, thus presenting a unique model system to assess the essential molecular determinants underlying mechanotransduction. Using cryo-electron microscopy and molecular dynamics simulations, we show that *Tc*MscS contains two short membrane-embedded helices that would not fully cross an intact lipid bilayer. Consequently, drastic membrane deformation is induced at the protein-lipid interface, resulting in a funnel-shaped bilayer surrounding the channel. Resident lipids within the central pore lumen block ion permeation pathway, and their departure driven by lateral membrane tension is required for ion conduction. Together with electrophysiology and mutagenesis studies, our results support a direct lipid-mediated mechanical gating transition. Moreover, these findings provide a foundation for the development of alternative treatment of Chagas disease by inhibition of the *Tc*MscS channel.

Protozoan pathogens experience drastically different osmotic and mechanical environments during transmission between hosts and migration from extracellular to intracellular stages[1–5]. Activation of mechanosensitive (MS) channels and subsequent initiation of downstream biochemical signaling are often necessary to cope with these dynamic mechanical stresses[6,7]. *Trypanosoma cruzi*, the causative agent of Chagas disease, encodes a mechanosensitive channel of small conductance (MscS) that is essential for parasite development and infectivity[8]. Chagas disease affects millions of people in central and South America and potentially causes irreversible chronic damage to

the heart, and digestive and nervous systems if left untreated[9,10]. Current therapeutic agents, including benznidazole and nifurtimox, exhibit limited efficacy with serious side effects[11–14]. Therefore, inhibition of *T. cruzi* MscS (*Tc*MscS) may represent an alternative strategy for the treatment of Chagas disease.

*Tc*MscS belongs to the superfamily of MscS-like channels that are ubiquitously found in bacteria, protists, fungi and plants, but not in animals[15–18]. The prototypical *E. coli* MscS (*Ec*MscS) channel has served as the prevailing model system for structural and functional analyses, advancing our understanding of the physicochemical principles

[1]Department of Pharmacological Sciences, Icahn School of Medicine at Mount Sinai, New York, NY, USA. [2]Department of Neuroscience, Icahn School of Medicine at Mount Sinai, New York, NY, USA. [3]Department of Biotechnology and Pharmaceutical Sciences, Western University of Health Sciences, Pomona, CA, USA. [4]WesternU Molecular Computing Core, Western University of Health Sciences, Pomona, CA, USA. [5]Department of Cell Biology and Physiology, Washington University School of Medicine, Saint Louis, MO, USA. [6]Center for the Investigation of Membrane Excitability Diseases, Washington University School of Medicine, Saint Louis, MO, USA. [7]These authors contributed equally: Jingying Zhang, Aashish Bhatt, Grigory Maksaev. ✉e-mail: luoy@westernu.edu; peng.yuan@mssm.edu

underlying mechanotransduction[19–31]. These studies have established that the surrounding lipid membrane plays a critical role in the mechanotransduction process, reinforcing the 'force-from-lipids' concept[23,28,29,31–34]. However, the mechanism by which lateral membrane tension exerts force from lipids to induce channel activation remains incompletely understood. Recent studies of MscS-like channels, including MscK[35], MSL1[36,37], and MSL10/FLYC1[38,39], have further increased our knowledge of mechanotransduction in these channels. Remarkably, an intrinsically curved transmembrane domain (TMD) is apparently present in MSL1 and MscK channels and appears to undergo large flattening and expansion upon channel opening[35,36]. This gating behavior seems to extend to the animal Piezo channels that are structurally and evolutionarily unrelated but share the structural feature of an inherently curved TMD[40–42], which undergoes flattening in liposomes as observed by cryo-electron microscopy (cryo-EM) and high-speed atomic-force microscopy (HS-AFM)[43,44]. By contrast, all the available structures of MSL10/FLYC1 indicate a rather flat TMD[38,39], and channel gating seems to involve only subtle conformational changes, such as side-chain rearrangement alone. Therefore, these MscS-like channels, while all sharing the MscS fold, could have divergent gating transitions, further underscoring the complex interplay between MS channels and their surrounding lipid membranes.

Amongst all the MscS-like channels identified and verified to date, TcMscS is unique at the molecular level in that it represents a minimal functional unit that maintains MS channel activity[8]. Each TcMscS subunit contains only two presumed transmembrane (TM) helices followed by a small soluble domain, which corresponds to a portion of that of the prototypical EcMscS (Fig. 1a). In particular, TcMscS lacks the equivalent first TM helix (TM1) and an N-terminal helix in EcMscS, which anchors to the outer leaflet of the lipid bilayer and appears to be critical for perception of membrane tension[23]. Moreover, TcMscS lacks the C-terminal portion that generates the cytoplasmic side portals essential for ion selectivity and passage[26,37]. These contrasting aspects render TcMscS a unique model system to interrogate the essential molecular and structural components underlying mechanotransduction in ion channel proteins. By combining electrophysiology, single-particle cryo-EM, and all-atom molecular dynamics (MD) simulations, we show that TcMscS forms a heptameric channel with a central ion conduction path, but indeed lacks the side portals shared by other MscS-like channels. Strikingly, the presumed two TM helices are of insufficient length to fully cross an intact planar lipid membrane and thus induce drastic membrane deformation surrounding the channel. Resident lipids within the central pore lumen block ion conduction in the resting state, and displacement of these lipids driven by increased membrane tension is required for ion conduction. These unusual structural arrangements of an ion channel with its surrounding lipid membrane support a direct lipid-mediated mechanical gating transition without apparent protein conformational changes. Furthermore, our findings provide a foundation for therapeutic development targeting TcMscS because of its essential role in parasite infectivity and the absence of homologous channels in humans.

## Results and discussion
### Cryo-EM structure of TcMscS

The full-length TcMscS, consisting of 165 amino acids, was heterologously expressed in *Pichia pastoris*, purified to homogeneity in glyco-diosgenin (GDN) detergents, and subjected to single-particle cryo-EM analysis. The 3D reconstruction reached an overall resolution of 3.21 Å with applied C7 symmetry (Fig. 1b and Supplementary Fig. 1). The AlphaFold prediction of a protomer was used as the initial model to facilitate model building, and the final atomic model was refined to good stereochemistry (Supplementary Table 1). Side-chain densities were well resolved for most of the amino acids and thus allowed unambiguous placement in the final model (Supplementary Fig. 2). The C-terminal region, including amino acids 138–165, was

not resolved in the cryo-EM density map and thus not included in the atomic model.

The heptameric TcMscS assembly forms a minimal channel architecture corresponding to the central core of other MscS homologs such as EcMscS[19] (Fig. 1c–e), EcMscK[35], and AtMSL1[36,37]. Each subunit consists of an N-terminal short helix (N helix) and two membrane-embedded helices, TM2 and TM3, followed by a β-domain on the cytoplasmic side (Fig. 1a). Like other MscS homologs, a pronounced kink separates the pore-lining helix to TM3a and TM3b (Fig. 1d). As indicated by its amino acid sequence and in contrast with all other MscS-like channels, TcMscS lacks the C-terminal α/β domain that generates the cytoplasmic side portals constituting an integral part of the ion conduction path (Fig. 1e and Supplementary Fig. 3). Concomitantly, the central pore formed by TM3 and the β-domain defines the complete ion permeation pathway in TcMscS.

Structural alignment of TcMscS and EcMscS indicates that the TcMscS channel has an unusual arrangement of its TMD within the lipid membrane (Fig. 1e). Given the membrane boundaries deduced from the full-length EcMscS structures[23,30], the TMD of TcMscS is apparently incapable of spanning a planar biological membrane. Even with the previously assumed membrane boundaries[19,20,29] (shifting downward by 14 Å), the charge distribution of the TM helices indicates that the hydrophobic thickness of the TMD of TcMscS is much reduced compared with that of EcMscS. This indicates that the membrane-embedded helices of TcMscS would not be able to cross an intact biological membrane. Moreover, compared with EcMscS, TcMscS lacks the long TM1 helix and the N-terminal membrane-anchoring helix, which defines the upper leaflet boundary of the membrane bilayer[23,30]. The thickness of the TMD of TcMscS was estimated to be ~ 24 Å, the distance measured between the Cα atoms of the charged residue R26 in TM2 and the outmost extracellular residue I44 (Supplementary Fig. 4a, b). Thus, the short length of TM2-3 in TcMscS indicates drastic membrane deformation at the channel-lipid interface (Fig. 1e and Supplementary Fig. 4a, b). Together, these structural observations and comparisons hint at an unusual arrangement of TcMscS within the surrounding membrane.

### The N helix

In contrast to other MscS-like channels, TcMscS contains a unique N-terminal short helix (N helix), which corresponds to only a fraction of TM1 in EcMscS (Fig. 1a, e and Supplementary Figs. 3, 4). The short length of the N helix of TcMscS makes it impossible to span an entire membrane. Moreover, several positively charged amino acids, K2, R3 and R7, are located within the N helix (Supplementary Figs. 3, 4), suggesting that the N helix is energetically unfavorable to be embedded within a lipid membrane. Instead, these positively charged amino acids appear to be positioned to interact with the head groups of typically negatively charged phospholipids in the inner leaflet. The carried charges of the N helix, along with its peripheral location in the channel assembly, imply that the N helix may be positioned at the lipid-water interface to primarily interact with surrounding lipids and thus sense the mechanical state of the membrane or facilitate anchoring of the channel to the membrane.

### Membrane deformation observed in MD simulations

The unusually thin TMD of TcMscS raises an intriguing question. How is the channel embedded in a regular lipid bilayer? To address this, we performed all-atom molecular dynamics (MD) simulations in three different types of bilayers with varying thickness and charge: neutral POPC (~ 36 Å), neutral DLPC (~ 28 Å), and DLPC with 20% anionic DLPA (~ 28 Å) (measured by the distances between phosphorous atoms in the outer and inner leaflets, Supplementary Table 2). Simulations were initiated by placing the channel at different z-positions relative to the membrane center. In all three bilayers, the TcMscS channel converged within 100 ns to the same equilibrium position, wherein TM2 and

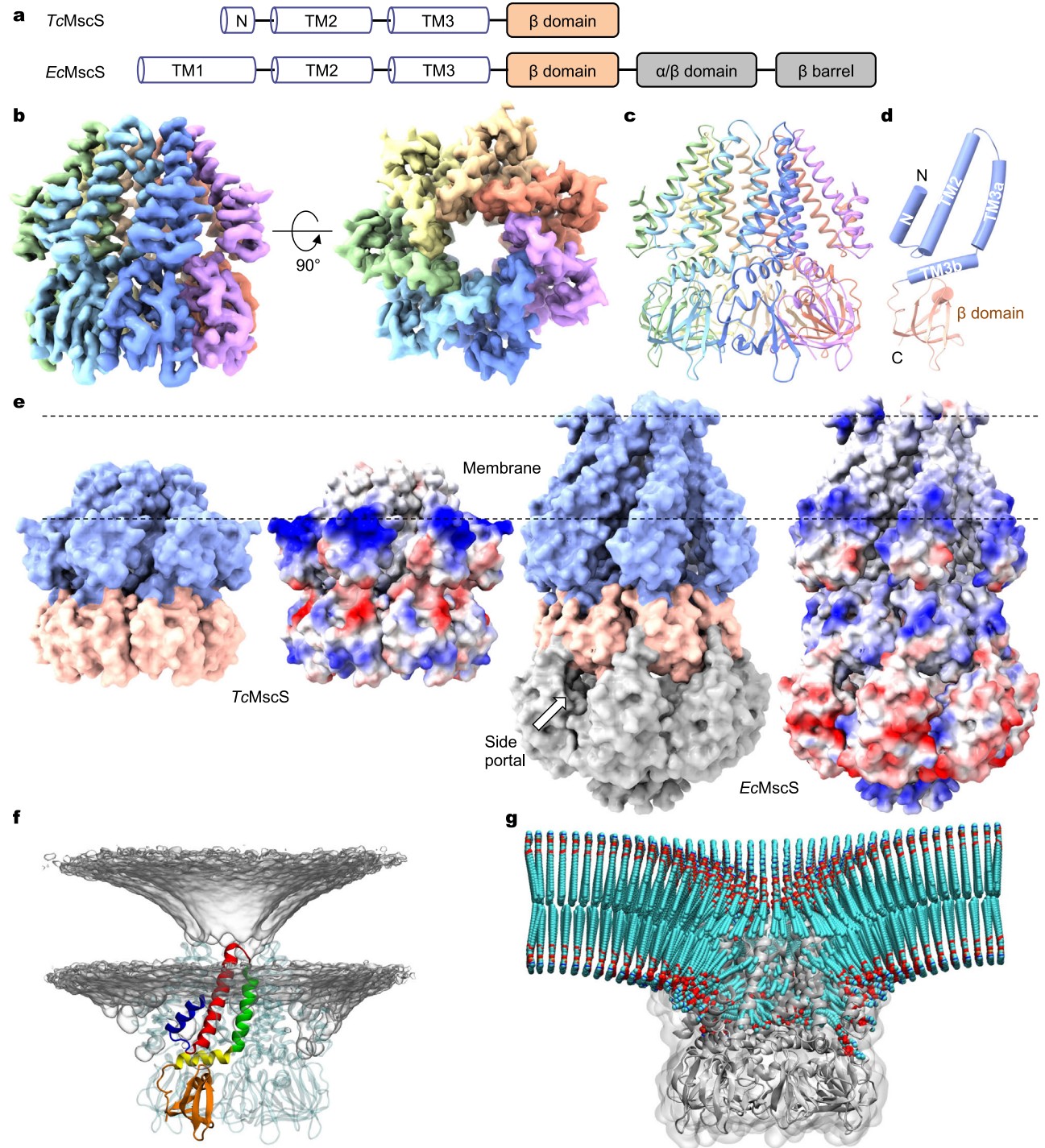

**Fig. 1 | Cryo-EM structure of *Tc*MscS. a** Domain organization of *Tc*MscS and *Ec*MscS. **b** Cryo-EM reconstruction of the heptameric *Tc*MscS channel. Shown are orthogonal views of the cryo-EM density map sharpened by deepEMhancer. Each subunit is in a unique color. **c** The overall structure. **d** Structure of a single subunit with each domain distinctly colored. **e** Surface representation (left: space-filling; right: electrostatic potential (red, − 10 kT/e; white, neutral; blue, +10 kT/e) of *Tc*MscS and *Ec*MscS (PDB: 6RLD). The structures of *Tc*MscS and *Ec*MscS were aligned by the conserved soluble β-domain. The dashed lines demarcate the membrane boundaries indicated by cryo-EM structures of the full-length *Ec*MscS. **f** All-atom MD equilibrated position of the wild-type *Tc*MscS in a POPC bilayer. Membrane deformation is depicted by the density of lipid headgroups shown in gray. A single-channel subunit is colored. **g** Lipid coordinates averaged over 2 μs trajectory using MOSAICS[76]. Lipids are colored by the atom types (cyan carbon, red oxygen, blue nitrogen, and orange phosphorus). The channel is shown in gray.

TM3a are partially embedded in the bilayers and the N helices are anchored below (Fig. 1f and Supplementary Fig. 5a). Deletion of the entire N helix in silico destabilized partitioning of the channel into the membrane and loosened TM2-TM3 packing, underscoring the critical role of the N helix in maintaining protein-lipid interactions (Supplementary Fig. 5b).

Remarkably, at the equilibrium position, *Tc*MscS deforms the surrounding bilayer into a prominent funnel shape, allowing the nearby lipids to accommodate its thin TMD (Fig. 1f). In addition, by occupying mainly the inner leaflet, *Tc*MscS appears to generate a membrane perforation in the outer leaflet. To understand how *Tc*MscS deforms the membrane to such a degree, we computed time-averaged

lipid coordinates over 2 µs of all-atom MD trajectories. The crevices between neighboring subunits facilitated downward movement of the inner leaflet lipids towards the β-domain on the cytoplasmic side (Fig. 1g). Consequently, lipids from the outer leaflet were pulled downwards. The redistribution of surrounding lipids was consistently observed for all three lipid compositions simulated, but not in the systems with the N helix of *Tc*MscS deleted (Supplementary Fig. 5b).

## Channel activation and inactivation

To examine its channel activity, *Tc*MscS was heterologously expressed in giant spheroplasts from *E. coli* strain MJF516[45], which lacks four endogenous MS channel genes (*mscs-, msck-, ybio-, yjep-*). Mechanosensitive ionic currents were measured from excised membrane patches under pressure ramps. The endogenously expressed *Ec*MscL channel, which opens at a high-tension threshold, provided an internal calibration of tension in the excised membrane patches. *Tc*MscS was activated by increased negative pressure in the recording pipette (Fig. 2a), with a unitary conductance of $478 \pm 11$ pS, which is consistent with a previous report[8]. Akin to *Ec*MscS[21] and *Vibrio cholerae* MscS (*Vc*MscS)[46], *Tc*MscS transitioned into an inactivated state following activation by several rounds of repetitive pressure ramps or constantly applied pressure (Fig. 2a, b). Intriguingly, recovery from inactivation was not observed within several minutes, with or without further increase of pressure (Fig. 2a, b). By contrast, previous studies have demonstrated that both *Ec*MscS and *Vc*MscS recover from inactivation within seconds following tension release and that amino acids near the TM3 kink in *Ec*MscS influence the rates of inactivation and recovery[21,46]. Specifically, the *Ec*MscS G113A mutant exhibited slower inactivation and faster recovery, whereas Q112G displayed a similar inactivation rate as that of the wild-type channel but with slower recovery[21].

Sequence alignment indicates variation in the kink region amongst these channels (Supplementary Fig. 3). To examine the roles of these amino acids in *Tc*MscS channel function, we generated chimeric constructs replacing residues 66-71 of *Tc*MscS with the corresponding region of *Ec*MscS (*Tc*MscS-*Ec*) or *Vc*MscS (*Tc*MscS-*Vc*). The *Tc*MscS-*Ec* chimera displayed flickery activity, with slower inactivation and unitary conductance being slightly lower than that of the wild-type *Tc*MscS (Supplementary Fig. 6a, b). The *Tc*MscS-*Vc* chimera also exhibited flickery openings with single-channel conductance similar to that of the wild-type *Tc*MscS (Supplementary Fig. 6a, c, d). However, in contrast to fast inactivation of the wild-type *Tc*MscS under sustained membrane tension, the *Tc*MscS-*Vc* chimera showed no inactivation (Supplementary Fig. 6c). Further examination of these TM3 kink regions allowed us to identify a point mutation, *Tc*MscS C66L, that demonstrated higher tension sensitivity (Supplementary Fig. 6d) and sufficiently abolished channel inactivation in excised patches (Fig. 2c). The distinct inactivation processes occurred in the wild-type and mutant channels motivated us to also determine the cryo-EM structure of *Tc*MscS C66L, which, surprisingly, was nearly identical to that of the wild type (root-mean-square deviations (RMSD) of ~0.4 Å for all Cα atoms, Fig. 2d). Thus, the structural basis underlying channel inactivation in *Tc*MscS remains unclear. Nonetheless, these results demonstrate that the TM3 kink region in these MscS homologs critically regulates rates of channel inactivation and recovery. Moreover, the identical structures of the wild-type channel and a non-inactivating mutant, C66L, indicate that our cryo-EM structures unlikely represent an inactivated conformation.

## Closed conformation mediated by pore lipids

To assess the corresponding functional state of the wild-type *Tc*MscS structure, we compared it with the closed and open *Ec*MscS conformations. Alignment of the conserved β domains in these structures indicates that the arrangement of TM2 and TM3 in *Tc*MscS is similar to that in the open *Ec*MscS channel but differs substantially from that in the closed (Fig. 3a), suggesting that the *Tc*MscS structure may

represent an open conformation and that *Tc*MscS adopts a membrane-embedded position closer to that of the open *Ec*MscS channel. Previous studies of *Ec*MscS have highlighted the essential role of lipids in stabilizing the closed conformation[23,28,31,47]. Specifically, the wild-type *Ec*MscS channel embedded in lipid nanodiscs or isolated with mild detergents maintained a closed conformation, but transitioned to an open conformation through removal of endogenously bound lipids by more dispersive detergents[28]. To compensate for the potential loss of lipids associated with *Tc*MscS during purification with detergents, we also reconstituted *Tc*MscS into lipid nanodiscs and determined the cryo-EM structure at a nominal resolution of 3.18 Å (Supplementary Fig. 1). The structures of *Tc*MscS in detergents and in nanodiscs are essentially identical, with an RMSD of ~0.3 Å for all Cα atoms (Fig. 3b). Thus, this conformation is inherently favored by the purified wild-type *Tc*MscS channel, either in detergent or lipid environment.

Comparison of the pore profiles of *Tc*MscS and closed and open *Ec*MscS channels indicates that the transmembrane region of *Tc*MscS has an intermediate opening (Fig. 3a, b), with the narrowest pore diameter of 10.8 Å at residue F64 (Fig. 3c, d). Many MscS-like channels contain consecutive bulky hydrophobic residues prior to the kink in the pore-lining helix, such as L105 and L109 in *Ec*MscS[20] (Fig. 3c), V319 and F323 in *At*MSL1[36], V549 and F553 in *At*MSL10[38], V568 and F572 in *Dm*MSL10/FLYC1[39], and V921 and F925 in *Ec*MscK[35], respectively. Structural and functional studies of these channels have demonstrated that the consecutive hydrophobic residues define the narrowest constriction and result in substantial dewetting along the central pore, thus functioning as a gate preventing water and ion permeation. By contrast, the equivalent region in *Tc*MscS (A60 and F64) contains a single bulky hydrophobic residue, F64, that forms the narrowest point (~5.4 Å in radius, Fig. 3c, d). Consequently, this results in reduced hydrophobicity at the presumed central gate of *Tc*MscS, thus lowering the energy barrier for water and ion conduction (Fig. 3e). Moreover, the peripheral N helix and TM2 of *Tc*MscS interact with the β domain from the same subunit, adopting a non-domain-swapped configuration, which is also consistent with that of the open, but not closed, *Ec*MscS conformation (Supplementary Fig. 4c). Together, these analyses suggest that the *Tc*MscS structure resembles the *Ec*MscS open conformation.

In contrast to all other known MscS-like channels, *Tc*MscS lacks the cytoplasmic cages that form part of the ion-conduction path (Fig. 1e), and thus its ion permeation pathway is completely defined by the central symmetry axis. On the cytoplasmic side, *Tc*MscS contains a unique β1-β2 loop, which harbors a hydrophobic residue, L97, that forms the narrowest constriction with a radius of ~3.8 Å (Fig. 3c, d and Supplementary Fig. 3). To evaluate the functional role of L97, we introduced a smaller-sized residue at this position, glycine (L97G), alanine (L97A) or valine (L97V). Maintaining the same gating pressure threshold as that of the wild type, all three mutant channels have an increased unitary conductance with the sequence of L97G > L97A > L97V > WT, which correlates inversely with the size of the side chain (Supplementary Fig. 6d). In contrast, introduction of a bulkier hydrophobic residue at this position (L97F) rendered a non-conductive channel (Supplementary Fig. 7), suggesting that the pore is physically blocked. These results indicate that L97 defines the unitary conductance but is not involved in mechanical gating, which is consistent with the notion that the cytoplasmic β-domains in MscS-like channels are rather rigid and remain static during channel gating[20,31,35,36].

The above structural analysis and comparison, focusing on the channel protein conformations alone, would suggest that the wild-type *Tc*MscS structures represent a conductive state. However, a wealth of lipid-like densities is associated with the channel, evident in the cryo-EM density map of *Tc*MscS in detergent or in lipid (Fig. 4a, b), which may interfere with ion conduction. Similar to the closed *Ec*MscS channel embedded in lipid nanodiscs[23,30,31,36], lipids are present at the inter-subunit interfaces, in the conserved lipid-binding pockets, and

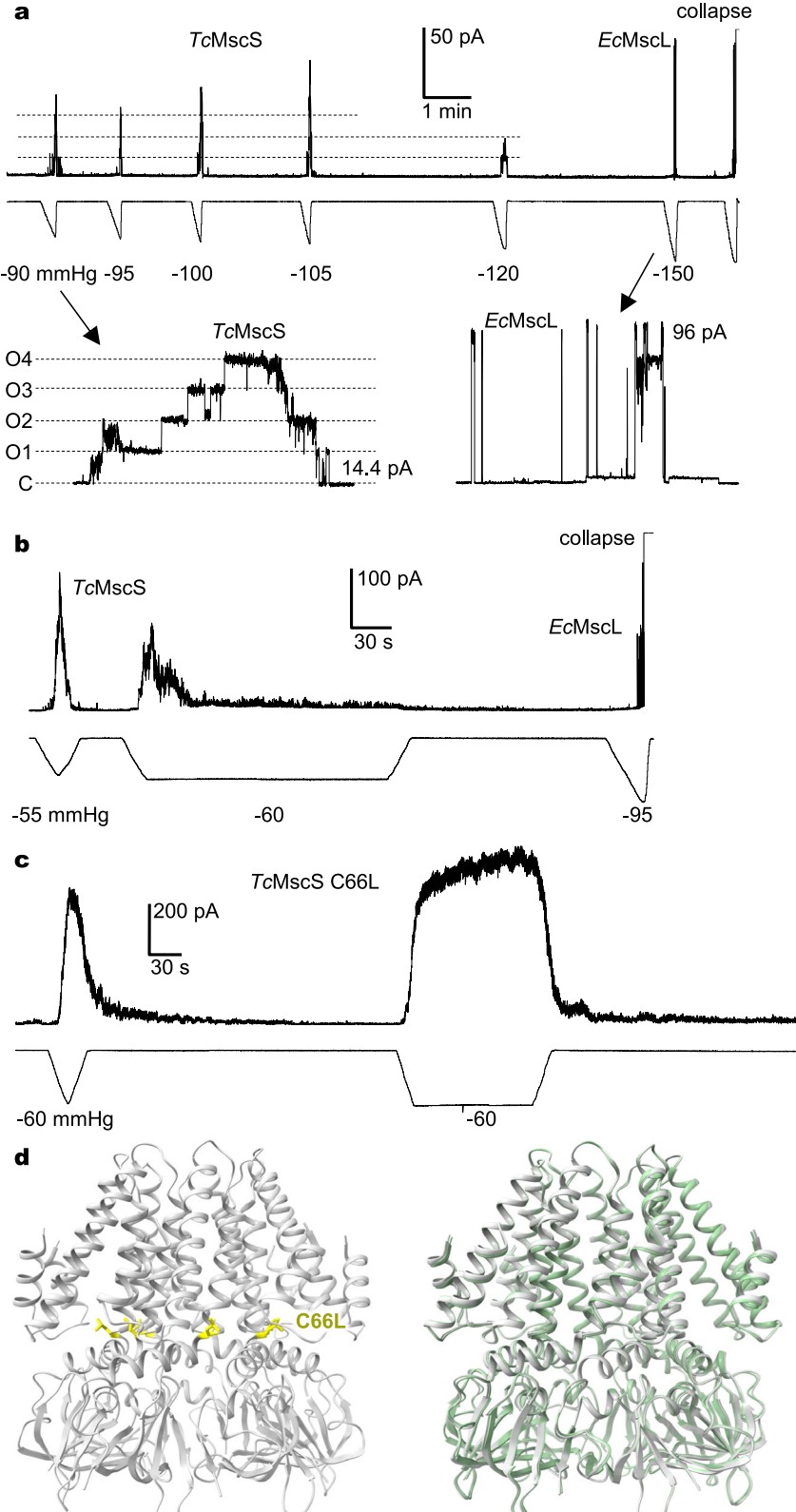

**Fig. 2 | Function of *Tc*MscS. a** *Tc*MscS inactivation in the absence of sustained tension. Over time, increased stimulus amplitudes failed to induce channel opening. Insets: opening and closure of *Tc*MscS channels in response to the first − 90 mmHg pressure ramp (left) and activation of endogenous *Ec*MscL channels, but not *Tc*MscS channels at the near-lytic − 150 mmHg pressure ramp (right). Symmetric KCl buffer (200 mM KCl, 90 mM MgCl₂, 2 mM CaCl₂, and 5 mM HEPES, pH 7.2) was used, and all measurements were carried out on excised inside-out patches at − 30 mV membrane potential. **b** Channel inactivation under sustained tension (− 60 mmHg pressure step). Subsequent near-lytic − 95 mmHg pressure ramp only induced activation of endogenous *Ec*MscL channels, but not *Tc*MscS channels. **c** *Tc*MscS C66L showed no significant inactivation either in the absence of tension or under sustained (− 60 mmHg step) tension. **d** Cryo-EM structure of *Tc*MscS C66L (in gray) and overlay with the wild-type *Tc*MscS (in green) structure.

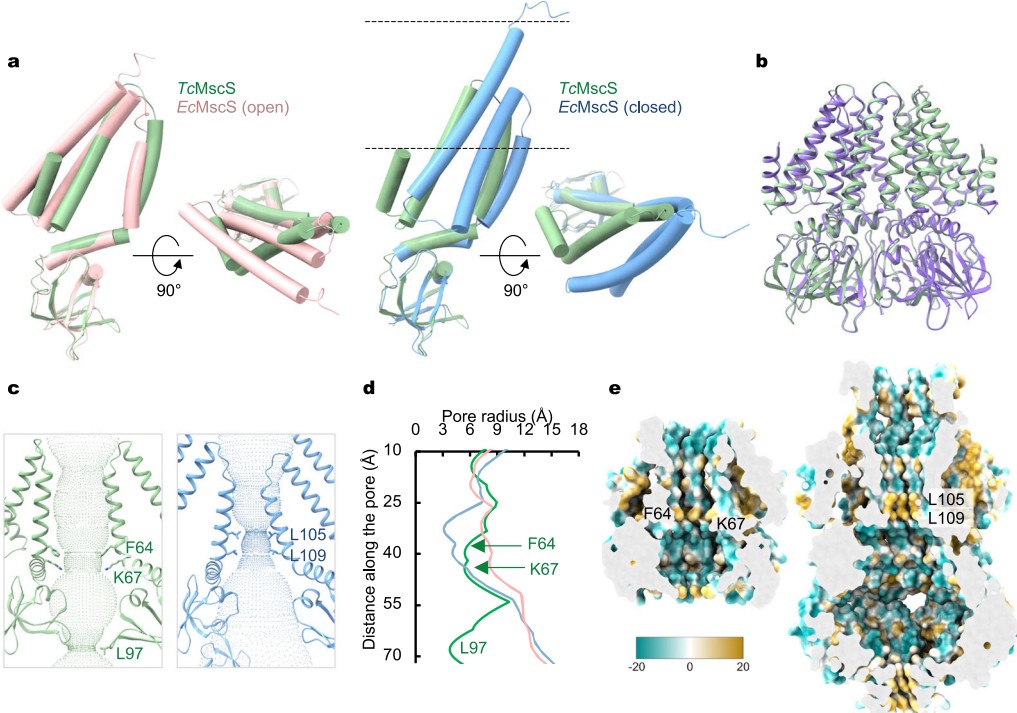

**Fig. 3 | Conformational state of *Tc*MscS. a** Superposition of the protomer structures of *Tc*MscS (green) and the open (pink, PDB: 2VV5) or closed (blue, PDB: 6RLD) *Ec*MscS channel. The dashed lines indicate the membrane boundaries. **b** Superposition of the wild-type *Tc*MscS structures in detergent and in lipid. **c** Ion conduction pores of *Tc*MscS (left panel, green) and closed *Ec*MscS (right panel, blue). **d** Comparison of the pore profiles of *Tc*MscS (green) with the closed (blue) and open conformations (pink) of *Ec*MscS. **e** Hydrophobicity of the central ion permeation pathway in *Tc*MscS (left panel) and *Ec*MscS (right panel).

within the pore lumen of *Tc*MscS (Fig. 4). Estimation of the pore radius with the inclusion of lipids in the atomic model rendered a hydrophobic region with much reduced opening, comparable to that of the closed *Ec*MscS channel (Fig. 4c). Therefore, it appears that the resident pore lipids occlude ion conduction, and that channel activation would require departure of the pore lipids from the central ion permeation path.

## Lipid blockade revealed by MD simulations

To further evaluate the conformational state and to investigate protein-lipid interactions in *Tc*MscS, we performed three replicas of 100 ns all-atom MD simulations with or without pore lipids (see Methods for system preparation). In the absence of lipids inside the pore, it remained fully hydrated, with a simulated unitary conductance of $393.6 \pm 33.1$ pS (Fig. 5a and Supplementary Fig. 5c), which is largely consistent with our measurements from excised membrane patches ($478 \pm 11$ pS) and previous studies[8]. In contrast, simulations with pore lipids showed seven POPC lipid tails within the pore, consistent with the observed lipid densities in the cryo-EM reconstructions, completely blocked water and ion permeation, regardless of the presence or absence of an applied 500 mV voltage (Fig. 5b and Supplementary Table 4). These results further support that our *Tc*MscS structures represent a lipid-occluded non-conducting state.

We then conducted all-atom MD simulations under a constant membrane tension at 10, 15, 20, or 32 mN/m. With applied tension, the funnel-shaped bilayer became thinner and nearly flat; however, inner leaflet lipids between neighboring subunits remained lodged in the crevices (Fig. 5c). Within 0.5 μs, we observed one to three lipid tails exiting the pore through the top groove formed by adjacent TM3a helices under one replica of 20 mN/m and in two replicas of 32 mN/m tension (Fig. 5d, Supplementary Movie 1, Supplementary Fig. 5d and Supplementary Table 4). However, these partial departures of pore lipids did not result in a conductive pore, as the remaining lipids

continued to obstruct the permeation pathway. Indicated by protein backbone RMSD and principal component analysis (PCA), no significant conformational changes were observed in the channel pore during 2 μs simulations (Supplementary Fig. 5d, e). Therefore, complete lipid departure may require longer simulation time or protein conformational changes.

These observations also led us to reason that D47, the only charged residue in the outer pore region, plays an important role in pore hydration and lipid residency (Supplementary Fig. 5f). To test this hypothesis, we computationally mutated all seven D47 residues to alanine. As expected, the D47A mutant rendered a more hydrophobic pore, with more lipid molecules occupying the pore (Supplementary Table 4). Inspired by these simulation results, we experimentally introduced the D47A mutation on the non-inactivating background C66L (D47A/C66L) and measured the activation pressure thresholds of C66L and D47A/C66L relative to that of the endogenous *Ec*MscL channel. Consistently, the D47A/C66L mutant channel maintained the same unitary conductance ($448.1 \pm 11.5$ pS, $N = 5$) as that of C66L, but was activated at a higher pressure threshold ($P_{MscL}/P_{D47A/C66L} = 1.95 \pm 0.51$, $N = 8$) than C66L ($P_{MscL}/P_{C66L} = 3.70 \pm 1.24$, $N = 8$).

While the 'force-from-lipids' concept in MS channels has been well established[23,28,29,31–34], the mode by which MS channels sense force from lipids remains a central question in understanding mechanotransduction. Recent progress in structural studies of multiple MS channels has clearly established the intimate and critical interactions between MS channels and their surrounding lipid environments[23,28,31,35,36,47,48]. In particular, large membrane deformation induced by the non-planar shape of the TMD of an MS channel is a common feature supporting a prevalent mechanical gating transition[35,36,40–43]. The extent of membrane deformation varies amongst these channels and appears to dictate the degree of structural rearrangement required for channel opening and thus tension sensitivity.

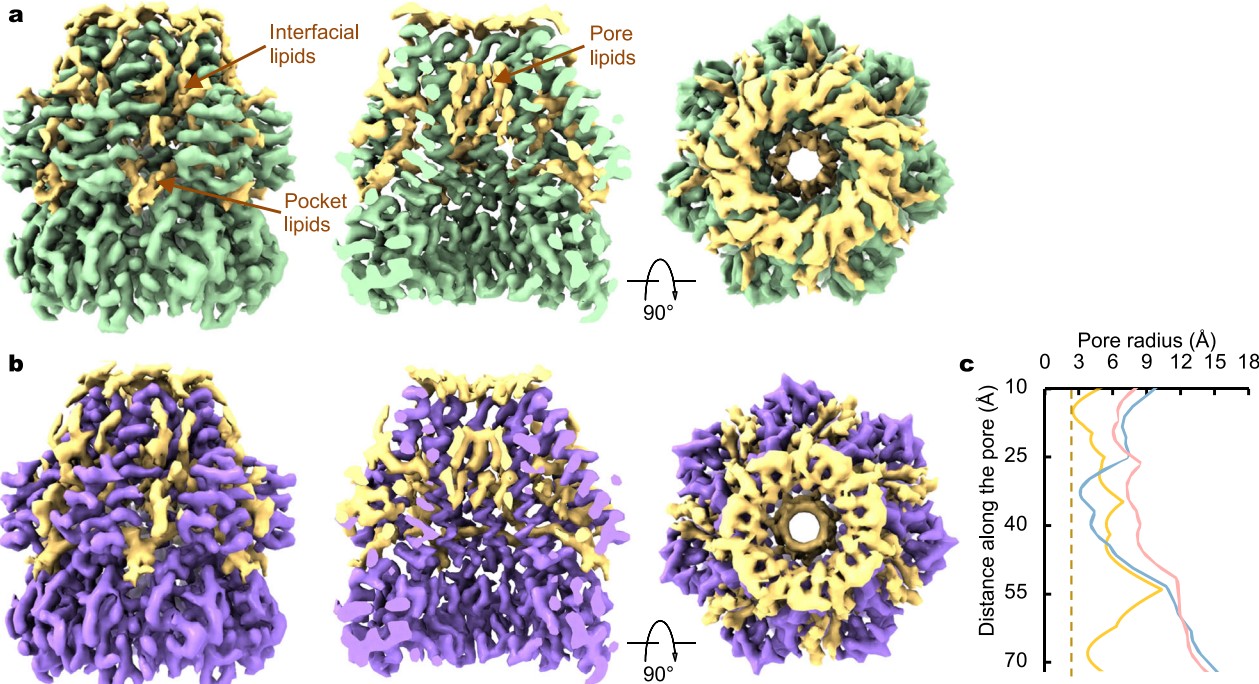

**Fig. 4 | Pore-blocking lipids. a, b** Lipid-like densities in the cryo-EM density map of the wild-type *Tc*MscS in detergents (**a**) or in lipid nanodiscs (**b**). Cutaway views in the middle panels highlight pore lipid densities. **c** The pore profile of *Tc*MscS with modeled lipids (in yellow), in comparison with the pore profiles of the closed (blue) and open *Ec*MscS conformations.

Here, we have shown drastic membrane deformation induced by a miniature MscS channel from *T. cruzi* with only two short membrane-embedded helices. The magnitude of membrane deformation caused by insertion of the channel into a bilayer is extraordinary because of the thin TMD of *Tc*MscS. The surrounding lipid bilayer is remodeled into a funnel shape, facilitated by the unique N helix of *Tc*MscS, which anchors the channel in the inner leaflet. Moreover, cryo-EM structures and MD simulations have demonstrated direct involvement of pore lipids in channel gating. Although partial delipidation in the pore was observed in multiple MD simulations under tension, a complete transition from the lipid-occluded nonconducting state to a conductive state was not captured within the short timescale of our simulations. In addition, the competition between lipids and water molecules inside the channel pore may be sensitive to the force field and thus requires more thorough investigation[49]. Further studies would be necessary to evaluate lipid-mediated gating of *Tc*MscS in biological membranes, especially in native membranes of *T. cruzi*.

## Methods

### Protein expression and purification

The *Tc*MscS gene (GenBank: KAF8291389.1) was codon optimized for expression in yeast cells and inserted into pPICZ-B vector with a C-terminal GFP-His$_{10}$ tag separated by a PreScission protease cleavage site. The plasmid was transformed into *Pichia pastoris* cells (strain SMD1163H, Invitrogen) for protein expression. Inverse PCR was used to generate mutations with the primers listed in Supplementary Table 3.

The wild-type *Tc*MscS and mutants were purified according to the membrane preparation protocol with some modifications[35,38,39]. Yeast cells were milled (Retsch MM400) for five times and resuspended in buffer containing 50 mM Tris-HCl pH 8.0 and 150 mM NaCl supplemented with DNase I (D-300-1, GoldBio) and protease inhibitors (3 µg ml$^{-1}$ aprotinin (A-655-100, GoldBio), 1 mM benzamidine (B-050-100, GoldBio), 100 µg ml$^{-1}$ 4-(2-Aminoethyl) benzenesulfonyl fluoride hydrochloride (A-540-10, GoldBio), 2.5 µg ml$^{-1}$ leupeptin (L-010-100,

GoldBio), 1 µg ml$^{-1}$ pepstatin A (P-020-100, GoldBio), and 200 µM phenylmethane sulphonylfluoride (P-470-25, GoldBio)). All the purification procedures were performed at 4 °C if not stated otherwise. The cell mixtures were centrifuged at 2500 × *g* for 10 mins. The subsequent supernatant was centrifuged at 100,000 × *g* for 1 h. The cell membrane pellets were then resuspended and homogenized in 50 mM Tris-HCl, pH 8.0 and 150 mM NaCl supplemented with DNase I and protease inhibitors. To extract protein from the cell membrane, the cell mixtures were added with 1% (w/v) glyco-diosgenin (GDN, GDN101, Anatrace) and stirred for 2 h, followed by centrifugation at 30,000 *g* for 0.5 h. The anti-GFP nanobody bound Glutathione Sepharose® 4B resin (GE Healthcare Life Sciences) was incubated with the subsequent supernatant for 3 h, followed by washing with buffer containing 20 mM Tris-HCl, pH 8.0, 150 mM NaCl and 85 µM GDN. PreScission protease was then added to remove the GFP-His$_{10}$ tag overnight. The protein sample was collected and injected into a Superose 6 Increase 10/300 gel filtration column (GE Healthcare Life Sciences) equilibrated with 20 mM Tris-HCl, pH 8.0, 150 mM NaCl and 40 µM GDN. Fractions corresponding to the target protein were collected for cryo-EM freezing.

### *Tc*MscS nanodisc reconstitution

Soybean polar lipid extract (541602 C, Avanti Polar Lipids, Inc.) was dried under Argon and desiccated with vacuum overnight. The lipid was then solubilized to ~10 mM in buffer containing 20 mM Tris-HCl, pH 8.0, 150 mM NaCl and 14 mM DDM (DDM, D310, Anatrace) and sonicated right before use. The purified *Tc*MscS protein was concentrated to ~3.2 mg ml$^{-1}$ and mixed with scaffold protein MSP1E1 and soybean polar lipid extract at a molar ratio of 1:0.5:50, and incubated for 10 mins on ice. The protein mixture was subsequently incubated with Bio-beads SM-2 resin (1523920, Bio-Rad) at a final volume ratio of 7:1 overnight, and then centrifuged at 2500 × *g* for 1 min to remove the Bio-beads SM-2 resin before injection into the Superose 6 Increase 10/300 gel filtration column equilibrated with 20 mM Tris-HCl, pH 8.0 and 150 mM NaCl. Fractions corresponding to *Tc*MscS nanodiscs were collected and subjected to cryo-EM freezing.

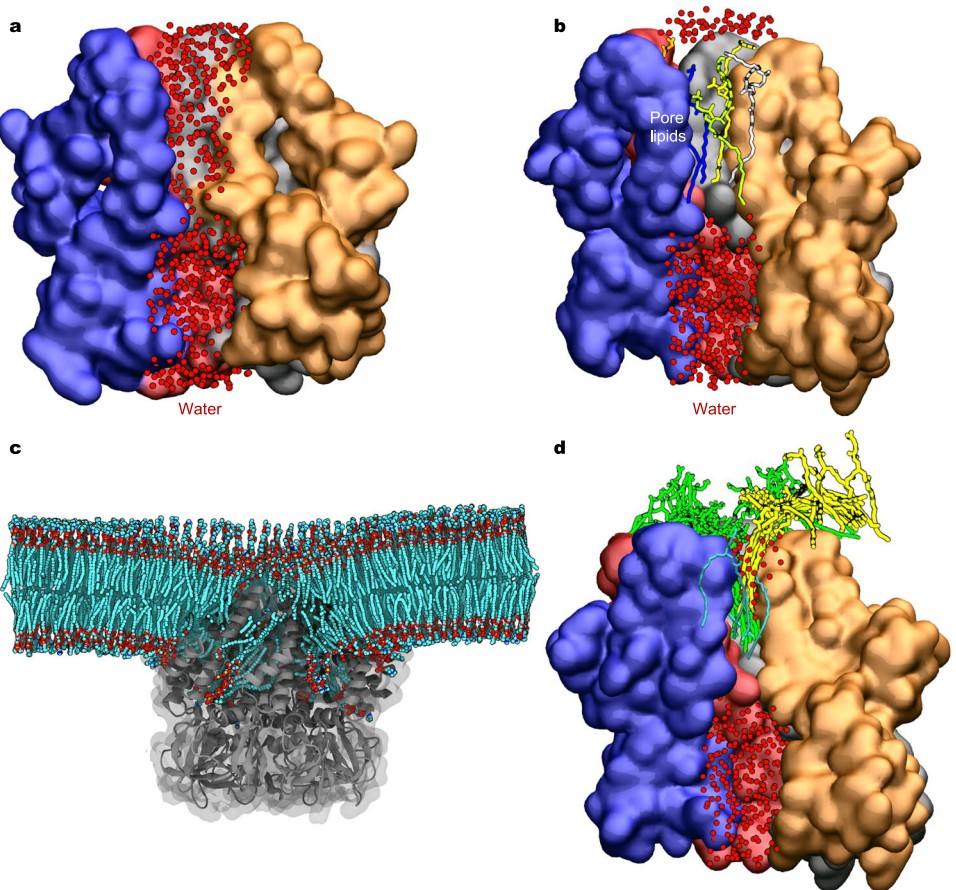

**Fig. 5 | All-atom molecular dynamics simulations of _Tc_MscS. a** MD simulation without pore lipids. The channel is shown in surface representation and colored by subunit. Front subunits are removed to show the central ion pore. Overlayed water molecules from a 100 ns trajectory are shown as red dots. **b** MD simulation with pore lipids. Pore lipids are shown in different colors using licorice representation. The ion pore is blocked by the pore lipids. **c** Flattening of the lipid bilayer with applied tension. Lipid coordinates were averaged over a 2 μs trajectory. Lipids are colored by the atom type: cyan carbon, red oxygen, blue nitrogen, and tan phosphorus. The channel is shown in gray. **d** Partial departure of pore lipids under 32 mN/m membrane tension (see Supplementary Movie 1 and Supplementary Fig. 5d).

## Cryo-EM sample preparation and data collection

Purified protein (3.5 μl wild-type _Tc_MscS at 7 mg ml⁻¹ in GDN detergent, 3.5 μl wild-type _Tc_MscS at 1 mg ml⁻¹ in nanodiscs, or 3 μl _Tc_MscS C66L at 2.15 mg ml⁻¹ in nanodiscs, respectively) was applied to the glow-discharged Quantifoil R1.2/1.3 Holey Carbon Grids (Q350CR1.3, Electron Microscopy Sciences). The grids for the wild-type _Tc_MscS in detergent or in nanodiscs were then blotted for 2 s after waiting for 20 s. For _Tc_MscS C66L in nanodiscs, the grids were blotted for 1.5 s after waiting for 8 s. The grids were then plunged into liquid ethane using FEI Vitrobot Mark IV (FEI). For the grids of the wild-type _Tc_MscS in detergent or in nanodiscs, images were collected using 200 kV Glacios Cryo-TEM with a Falcon4 detector (ThermoFisher Scientific) at a magnification of 120 k with a raw pixel size of 1.2 Å. For the grids of _Tc_MscS C66L in nanodiscs, images were collected using 300 kV Titan Krios Cryo-TEM with a K3 detector (Gatan, Inc.) at a magnification of 81 k with a raw pixel size of 1.083 Å. The defocus range was from − 0.8 to − 2.4 μm. Dose-fractionated images were recorded with a total dose of 43.11, 42.46, and 54.33 electrons per Å² per second for the wild-type _Tc_MscS in detergent, in nanodiscs, and _Tc_MscS C66L in nanodiscs, respectively.

## Cryo-EM data processing

For the wild-type _Tc_MscS in detergent, in nanodiscs, and _Tc_MscS C66L in nanodiscs, 3269, 3956 and 5985 movies were imported into cryoSPARC V3.4.0, V3.3.1 and V2.15.0, respectively[50]. After patch motion correction and patch contrast transfer function (CTF) estimation, 3226, 3584 and 5586 good images, respectively, were manually selected for blob picking and template picking. 2193852, 1467083, and 2244568 particles, respectively, were auto-picked and subjected to reference-free 2D classification. Good particles showing different orientations of the channel were used to generate 3D ab-initio models, which were utilized as the templates for heterogeneous refinement without enforced symmetry. 229377, 182063 and 124880 final particles, respectively, containing high-resolution information of the entire channel, were subjected to non-uniform refinement and local refinement with imposed C7 symmetry, using masks generated from Chimera 1.14[51]. To further improve the map quality of _Tc_MscS in detergent to guide model building, half maps from local refinement were imported into DeepEMhancer[52] and post-processed at highRes mode.

## Model building and refinement

The AlphaFold[53] model of the _Tc_MscS monomer (AF-A0A2V2X819, https://alphafold.ebi.ac.uk/entry/A0A2V2X819) was docked into the EM density maps and then manually adjusted in COOT 0.9.6[54]. The final atomic models were refined to good stereochemistry using real-space refinement in Phenix 1.20.1[55] and evaluated with MolProbity[56]. HOLE 2.2.005[57] was used to analyze the pore radius. Structural illustrations were prepared using ChimeraX 1.5[58].

## Giant spheroplasts preparation

_E. coli_ giant spheroplasts were prepared from the MJF516 cell line (_msck-, mscs-, ybio-, yjep-_) using WT _Tc_MscS and its mutants in the pET300 vector following the protocol[45,59]. Briefly, filamentous cells were

produced by growing in Luria-Bertani (LB) medium containing carbenicillin and cephalexin for 1.5 h. Cells were further induced by 1 mM Isopropyl β-D-1-thiogalactopyranoside (IPTG) for 1 h and harvested by centrifugation. The pellets were gently resuspended in 1.25 ml 1 M sucrose. Lysozyme digestion was carried out for 15 minutes via addition of 75 μl 1 M Tris pH 8.0, 50 μl 5 mg ml⁻¹ lysozyme, 15 μl 5 mg ml⁻¹ DNAseI, and 50 μl 125 mM EDTA pH 7.8. The reaction was terminated by the addition of 0.5 ml of stop solution (875 μl 1 M sucrose, 125 μl water, 20 μl 1 M MgCl₂, and 10 μl 1 M Tris pH 8.0). Cell suspensions were added on top of 7 ml of ice-cold dilution solution (10 mM MgCl₂ and 10 mM Tris, pH 8.0 in 1 M sucrose) into the 10 ml glass tubes. Spheroplasts were isolated by centrifugation at 4 °C, gently resuspended in ~ 0.5 ml of dilution solution, aliquoted and stored at − 80 °C.

### Electrophysiology

Symmetric KCl buffer (200 mM KCl, 90 mM MgCl₂, 2 mM CaCl₂, and 5 mM HEPES, pH 7.2) was used in all patch-clamp experiments, with bath solution supplemented with 500 mM sucrose. All records were made from inside-out patches excised from the spheroplast membrane. To generate lateral tension in a membrane patch, suction was manually applied to the pipette via syringe. Recordings were made and digitized with the Axopatch 200B patch-clamp amplifier, the Digidata 1320 digitizer (Molecular Devices), and PM-015R pressure monitor (World Precision Instruments). Data were collected at 5 kHz, lowpass filtered at 2 kHz and analyzed with the pClamp software suite (Molecular Devices). Pipettes with ~ 1 MΩ resistance were fabricated from the Kimble Chase soda lime glass using a Sutter P-96 puller (Sutter Instruments). All measurements were carried out at − 30 mV membrane potential. Activation of mechanosensitive channels was induced by a slow ~ 4 mmHg s⁻¹ negative pressure ramp, applied to an excised patch.

Tension sensitivity of the WT $Tc$MscS and mutants was assayed using the ratio of $P_{McsL}/P_{MscS}$, where $P_{McsL}$ and $P_{MscS}$ are the pressures at which the first MscL and $Tc$MscS channels were activated. Endogenous $Ec$MscL was used as an internal reference. Higher $P_{McsL}/P_{MscS}$ ratios indicated higher tension sensitivity of a given $Tc$MscS construct with respect to $Ec$MscL. The measurements were averaged from multiple (at least three) patches expressing the WT $Tc$MscS or mutations.

### All-atom molecular dynamics simulation

All simulated systems were prepared using the CHARMM-GUI Membrane Builder server[60,61]. Because the thickness of the presumed transmembrane domain of $Tc$MscS is less than that of a typical lipid bilayer, we prepared multiple systems to ensure that channel–membrane interactions were not biased by the starting points. In these systems, the channel was either aligned using the Orientations of Proteins in Membranes (OPM) database[62] or manually placed within the membrane at multiple different z-positions. To explore the effect of bilayer thickness and charge on protein stability and function, three types of symmetric bilayers were used: a 1-palmitoyl-2-oleoylphosphatidylcholine (POPC) bilayer, a 1,2-dilauroyl-sn-glycero-3-phosphocholine (DLPC) bilayer, and a DLPC with 20% of anionic 1,2-Dilauroyl-sn-glycero-3-phosphate (DLPA) bilayer (Supplementary Table 2). The protein and bilayer systems are solvated by a 150 mM KCl solution. The AMBER2020 CUDA software package[63] was used for equilibrium and three replicas of sub-microsecond production runs for each system with the CHARMM36m force field[64,65] and TIP3P water model[66].

To prepare the membrane-embedded systems with and without pore lipids, two equilibrium protocols were used. In the first protocol, the system energy was minimized for 5000 steps using the steepest descent method[67], followed by sequential equilibration in which water molecules were relaxed first, with the application of lipid head group positional restraints for 125 ps and protein backbone restraint for 1 ns. This protocol yielded seven lipid tails inside the pore, in agreement with the lipid densities observed in cryo-EM reconstructions. In the second protocol, lipid head group restraints were extended for 1 ns to allow the pore being

fully hydrated first. This protocol generated a conducting pore without pore lipids in the wild type, but a lipid-occluded pore in the D47A mutant.

To determine whether each system has reached equilibrium, the RMSD of the protein backbone, the z-positions of the center of mass of the channel relative to the membrane center of mass, and the number of water and lipid tails in the pore are all monitored over time (Supplementary Fig. 5). An integration timestep of 2 fs was used, and the SHAKE algorithm was used for the constraint of hydrogen atoms. The Particle Mesh Ewald method is used for calculating long-range electrostatic interactions[68]. A12 Å cutoff was used for the short-range nonbonded interactions. Temperature and pressure were controlled by using the Monte-Carlo barostat[69,70] and Langevin thermostat[71,72] to maintain the pressure at 1 bar and temperature at 313.15 K, respectively.

Three 2 μs simulations under membrane tension were conducted using the ANTON2[73] supercomputer, following the protocol described in ref. 74. Briefly, Lennard-Jones interactions were truncated at 11–13 Å and long-range electrostatics were evaluated using the $k$-Gaussian Split Ewald method. A NPγT (constant normal pressure, lateral surface tension, and constant temperature) ensemble was applied in the membrane plane with surface tension of 10, 15, 20, or 32 mN/m. Pressure regulation was accomplished via the Martyna−Tobias−Klein barostat, with a tau (piston time constant) parameter of 0.0416667 ps and reference temperature of 313.15 K. The barostat period was set to the default value of 480 ps per timestep. Temperature control was accomplished via the Nosé−Hoover thermostat with the same tau parameter. The $mts$ parameter was set to four timesteps for the barostat control and one timestep for the temperature control. The thermostat interval was set to the default value of 24 ps per timestep.

Ionic conductance was computed from − 0.3, − 0.4, − 0.5 V voltage simulations using the AMBER2020 CUDA version (Supplementary Fig. 5). VMD[75], MOSAICS[76] and MDAnalysis python package[77] were used for data analysis and figure rendering.

### Reporting summary

Further information on research design is available in the Nature Portfolio Reporting Summary linked to this article.

## Data availability

The cryo-EM maps and atomic coordinates have been deposited to the Electron Microscopy Data Bank (accession codes: EMD-44520, EMD-44521, and EMD-44522) and Protein Data Bank (PDB entry codes: 9BGQ, 9BGS, 9BGT). The raw images have been deposited to the Electron Microscopy Public Image Archive (accession codes: EMPIAR-12056, EMPIAR-12057, and EMPIAR-12058). For molecular dynamics simulation data, initial coordinates, simulation input files, and a coordinate file of the final output are available at https://github.com/LynaLuo-Lab/TcMscS. All numerical data related to Supplementary Fig. 6d are provided as a Source Data file with this paper. Source data are provided in this paper.

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

## Acknowledgements

This work was supported by NIH grants GM143440 (to P.Y.) and GM130834 (to Y.L.L.). The computational resources were provided by the Advanced Cyberinfrastructure Coordination Ecosystem: Services & Support (ACCESS) program, which is supported by National Science Foundation grants #2138259, #2138286, #2138307, #2137603, and #2138296, and the Pittsburgh Supercomputing Center Anton2 allocation MCB170106P, which is supported by NIH Grant GM116961. We thank the staff scientists at Washington University Center for Cellular Imaging for data acquisition. Part of the cryo-EM work was performed at the Simons Electron Microscopy Center at the New York Structural Biology Center, with major support from the Simons Foundation (SF349247).

## Author contributions

J.Z. performed biochemical preparations, cryo-EM experiments, structural determination and analysis. A.B. performed molecular dynamics simulations. G.M. conducted electrophysiology experiments. P.Y. conceived and supervised the project. J.Z., A.B., G.M., Y.L. and P.Y. analyzed the results and prepared the manuscript. Correspondence and requests for materials should be addressed to Y.L. and P.Y.

## Competing interests

The authors declare no competing interests.
