## [Transparent Peer Review file · Nature Communications]

Lipid-mediated gating of a miniature mechanosensitive MscS channel from *Trypanosoma cruzi*

Corresponding Author: Dr Peng Yuan

Version 0:

Reviewer comments:

Reviewer #1

(Remarks to the Author)

This study investigates the mechanosensitive channel TcMscS from *T. cruzi*, a parasite responsible for Chagas disease, using cryo and MD simulations. Unlike its *E. coli* counterpart (EcMscS), TcMscS has a minimal structure but retains mechanosensitive activity, causing membrane deformation at the protein-lipid interface. The study finds that ion conduction requires lipid displacement rather than major conformational changes in the channel, offering insights into lipid-mediated gating caused by TcMscS. The computational work is reasonably performed, with sufficient detail in the methods section, standard protocol usage, and basic analysis performed in replicate. The overall paper is clear to read. My suggestions on major revision relate to manuscript improvement include better integration of computational and wet lab data, as to further connect the experiments performed, utilize better subheadings to classify the work done, and consider a few additional analysis metrics to confirm the lack of conformational change resulting in the gating. Comments below:

1. Improve the sub-heads to be completer and more directed at the outcomes of those sub sections.
2. Improve integration of wet lab work to confirm simulation outcomes/movement of lipids.
3. Discuss limitations of study design related to MD simulations – force field selection impact, time scales, etc.
4. Perform analysis to confirm a lack of structural change more clearly – for example – you note that “Under 20 mN/m tension, two lipid tails exited the pore through the gaps between neighboring TM3a 14 helices around 0.5 μ s, but the pore was still blocked by the remaining lipid tails, without significant conformational changes of the channel.” How have you classified significant or not significant conformational change? PCA analysis could be beneficial here, as well as structural overlays of different time points to show lack of change (as well as comparison to initial cryo structure). If doing structural overlays, please perform clustering over time periods as to not cherry pick structures.
5. More discussion and demonstration of trends among the three replicates of data – in the SI each simulation cluster structure comparison to initial starting state should be shown so the reader can see this lack of conformational change in a structure perspective in addition to figure SI 5.
6. In the methods, it would be helpful to indicate how the channel protein was placed into the membrane structure – what amino acids were used for anchoring at the leaflet interface to relate to experimental date for accurate placement. You should also include more details on how analysis was performed/packages/etc.,
7. Minor – in the methods, you do not have molarity associated with your ion concentration.
8. Data availability – please consider posting simulation input (config files, starting structure) and major (at least .pdb) output files onto Zenodo or other file sharing platforms.

Reviewer #2

(Remarks to the Author)

This manuscript reports exciting findings on TcMscS, a mechanosensing channel from *Trypanosoma cruzi* with significant therapeutic potential. The authors utilized a combination of structural, functional, and computational approaches to elucidate the structure, conformation, and gating mechanisms of TcMscS. Their results reveal unique structural features of TcMscS, dramatic membrane deformation induced by the channel, and evidence to support a lipid-mediated gating mechanism. These insights significantly advance our understanding of this important mechanosensing channel, provide novel insights into mechanosensing, and lay a foundation for the development of therapeutic strategies targeting *T. cruzi*. Overall, these findings are significant and novel, with high impact in the field.

Several comments that need to be addressed:

1. The conformational state of the reported structure needs clarification. The author propose that the channel was captured in an open state. However, they also suggested 'direct lipid-mediated mechanical gating transition without major conformational changes.' If this is the case, would the 'closed' state differ significantly from the proposed open state? Potentially, the observed pore-blocking lipids may actually suggest a closed state. In addition, the cryo-EM samples in nanodisc were prepared without applied tension, which may potentially favor a closed conformation. In any case, please further clarify the conformational state of the current structure.
2. Fig. 3a shows that TcMscS overlays well with the open conformation of EcMscS. Does this imply that TcMscS adopts a similar membrane-embedded position as the open conformation of EcMscS? Please comment on their membrane boundaries.
3. In the coarse-grained simulation, it seems that the protein conformation was constrained during the simulations, while the lipids were not. Please clarify the rationale and comment on whether this might impact the lipids' relative movement in relation to the protein.
4. What is the effect of mutating the positive charged residues in the N-helix?
5. The proposed competition between lipids and water is interesting. Under typical physiological conditions, is water always present, and how does this affect the competition?
6. Minor: In method section: 'The purified TcMscS protein was concentrated to ~3.2 mg ml⁻¹ and mixed with soybean polar lipid extract and scaffold protein MSP1E1 at a molar ratio of 1:0.5:50 for 10 mins'. Please clarify the molar ratio among protein:lipid:MSP.

Reviewer #3

(Remarks to the Author)

Reviewer #4

(Remarks to the Author)

This study on lipid-mediated gating of a miniature mechanosensitive MscS channel reports cryo-EM structures of this channel, complemented with functional and computational investigations. The authors found that the unusually short transmembrane domain of this MscS channel results in a drastic deformation of the membrane around the channel. They observed in the cryo-EM maps lipid-like densities in the pore and that the occupancies of the modelled lipids decreases as a function of lateral tension and pore hydration in molecular dynamics simulations. Based on these observations, the authors propose a direct lipid-mediated mechanical gating transition that might happen without major conformational changes of the channel protein. This study is of interest to the broad readership of Nature Communications.

Fig 2:

The authors' presented representative recordings but a more quantitative and statistically sound presentation is needed, i.e., quantification of observables with at least 3 biological replicates. In particular, the following would strengthen the authors' argument.

1. A pressure-response relation of TcMscS constructed using the authors' setup.
2. Statistical analysis of the 'inactivation' in the absence of sustained tension.
3. Statistical analysis of the inactivation during sustained tension of TcMscS and TcMscS-C66L.

Line 220-222, 'Moreover, the identical structures...inactivated conformation.':

The observation that a seemingly activating mutant does not change the conformation alone provides little support for suggesting or rejecting whether the determined structure is or is not in an inactivated conformation.

Line 268-270, 'In contrast, introduction of a bulkier hydrophobic residue...':

No data have been provided in support of this statement.

Line 270-273, 'These results indicate that L97 defines...':

The authors have not provided any data showing that L97 does not modify the pressure-response relation – an effect of L97 on mechanical gating would be manifested in a change in the conformational equilibrium under tension.

Line 282-285, 'Estimation of the pore radius...' and Fig 4b:

The lipid lined pore radius is almost 3 Angstroms and seems to correspond to where polar lipid headgroups are situated, which might be sufficient for the passage of hydrated physiological ions. Perhaps the authors can provide more description/explanation on the lipid gating mechanism.

Fig 4c:

The authors could overlay the pore radius of TcMscS without modelled lipids to show the difference that lipid occupancy introduces.

Line 316, 'These simulations suggest that the pore is conductive in the absence of pore lipids.':

Related to my comment above on lipid gating, perhaps the authors could perform the same voltage simulations to estimate the conduction properties of the lipid-lined state with the bound lipids restrained, which would provide further support to the lipid gating model.

Line 323-328, 'To corroborate the simulation results, we...':

Please provide the associated experimental data with at least 3 biological replicates.

Reviewer #5

(Remarks to the Author)

(Remarks to the Authors):

Using cryo-electron microscopy, electrophysiology and molecular dynamics simulations, Zhang and colleagues found that TcMscS (in detergent and nanodiscs) has two short membrane-embedded helices that do not fully span the lipid bilayer and a unique short N-terminal domain as well as a different pore architecture (ion permeation pathway), causing significant membrane deformation. This results in a funnel-shaped bilayer around the channel, where resident lipids block the ion permeation pathway. Ion conduction requires these lipids to be displaced by lateral membrane tension. Further, patch-clamp measurements and mutagenesis studies suggest that TcMscS gating is directly mediated by lipids, without major conformational changes in the protein. These findings provide insight into mechanosensitive channel function.

The proposed data are clearly novel, and the structure(s), for the first time, highlight unique structural features of the minimal orthologue of MscS channels, as detailed above. These findings are of great significance, given the involvement of this channel in the biological process of protozoan development and infectivity due to its mechanosensitive features. Importantly, the structural and functional data presented in this manuscript provides a structural framework for the development of specific drugs to target/inhibit TcMscS channels.

In my opinion, the methodology used, as well as data presentation and quality of the data seem very high, with all necessary (supporting) files provided to judge the quality of the data. Although I am not an expert on cryo-electron microscopy, the provided density maps and quality reports seem convincing. Overall, the manuscript is very well written with clear figures and is of great interest to the readership of Nat. Commun.

In light of the unique structural features of TcMscS and its biological importance, I would be happy to support the acceptance of this manuscript. However, some major concern (discussed below) related to the functional and MD approaches needs to be addressed.

1. The significance of the N-terminus has only been investigated through in-silico experiments (MD simulations). However, it is speculated that the positive charges attract the negative charges of the lipid head groups of the inner membrane leaflet, which in turn has a strong impact on the deformation of the bilayer. However, removing the entire N-terminal region could induce various conformational changes, which in turn could lead to a volume change and consequently deformation. Whether the effect is truly due to specific charge interactions could best be demonstrated by neutralizing the charges and measuring the response to mechanical stimuli. In my view, these electrophysiological measurements would strengthen the manuscript.

2. Another critical point is that TcMscS is resolved in the open state (in comparison to EcMscS), with the permeation pathway seemingly blocked by pore lipids, which exit to the extracellular funnel-shaped space due to lateral membrane tension. A direct demonstration that this clears the channel pore and allows K⁺ permeation would be to block the visible current, potentially also in the non-inactivating C66L mutant. Additionally, how can the observed desensitization to mechanical stimuli be explained in the context of this lipid-induced pore plug mechanism?

3. Another point concerns the competition between pore lipids and water within the channel pore. While the difference from E. coli MscS channels has been described, indicating that the TcMscS pore is less hydrophobic but still hydrophobic enough to exclude water, it remains unclear how this competition could arise. Does the description of water refer to the pore from the intracellular side up to F64 or water in the extracellular pore?

4. The figures could benefit from additional information and details. While most details are provided in the legends, some are only mentioned in the methods section, making the visual representation not always self-explanatory. For example, in Figure 2, it is unclear what currents are being measured, at what potential, and which ion species are involved. Additionally, some details appear contradictory. Figure 2a does not clearly indicate that TcMscS transitions into an inactivated state upon repetitive pulses. Instead, a high negative pressure (-120 mmHg) seems to result in reduced activation. A recording showing repetitive pulses at the same pressure, where channel activity decreases over time, appears to be missing. Furthermore, the endogenous reference channel is described as having a high activation threshold compared to TcMscS. However, in Figure 2b, it appears to be activated at -95 mmHg, the same pressure that produces robust TcMscS activation in Figure 2a. How

can mixed activity be ruled out in this case? Was a calibration performed, for example, by heterologously expressing a non-functional TcMscS channel (dominant-negative mutant) in the cell line and determining the pressure profile of the endogenous EcMscS?

Reviewer #6

(Remarks to the Author)

In the manuscript "Lipid-mediated gating of a miniature mechanosensitive MscS channel from *Trypanosoma cruzi*" by Zhang et al the authors collected structural data for TcMscS, used molecular dynamic simulations to uncover the channel gating dynamics. This manuscript gives a deep insight into the gating structure of TcMscS through new structures, new models that observe lipid movement during channel gating, and patch clamp electrophysiology where single channels were observed. In general I found the manuscript to be well written and of interest to the broad scientific community as well as the mechanosensitive ion channel community. The methods section was clear and would allow for other scientists to complete similar work. I do have several major and minor corrections as outlined below.

Major

Patch clamp electrophysiology of point mutations and chimeras:

-This data is interesting but incomplete. In some cases the pressure threshold is reported for some mutations but not all mutations (lines 326-328). This value is critical in the comparison of point mutations to the wild type. Additionally, you report the unitary conductance for some mutations but not all can you please include this value for all point mutations, the wild type, and any chimeras that you discuss in the manuscript.

-For each mutation that you created and discuss in the manuscript can you show a representative trace?

-In line 269 you report that L97F does not conduct in response to applied tension, this would implies that the channel is surface expressed. Can you report how you confirmed that this mutant was surface expressed? Additionally, can you show a representative trace for the data collected?

-in Figure 2 panel b and c it appears that the channels display continued activity after the tension is removed, does this suggest that the channel has difficulty closing if the tension is removed prior to the channel entering the desensitized state?

Lines 326-328: You reference the pressure threshold for two mutants without inclusion of the WT channel threshold, this makes it difficult to compare the change in threshold to the WT channel. Can you please include the pressure threshold for this channel and representative traces for these mutations?

Extended Data Figure 6: Please include unitary conductance values for all mutations studied in the manuscript.

Minor

Line 473:update the the text to show how threshold values were calculated for all mutants

Version 1:

Reviewer comments:

Reviewer #1

(Remarks to the Author)

The authors have addressed concerns.

Reviewer #2

(Remarks to the Author)

The authors have adequately addressed my comments. The manuscript is ready for publication.

Reviewer #4

(Remarks to the Author)

The authors have addressed the major concerns I raised.

Reviewer #5

(Remarks to the Author)

My points have all been satisfactorily addressed or taken into account. Congratulations!

Reviewer #6

(Remarks to the Author)

The authors have satisfactorily addressed all of my comments, no additional edits need to be made to address my concerns.

Thank you for including the additional patch clamp data.

We thank all the reviewers for their positive and constructive comments. Below we address each comment raised by the reviewers.

REVIEWER COMMENTS

Reviewer #1 (Remarks to the Author):

This study investigates the mechanosensitive channel TcMscS from *T. cruzi*, a parasite responsible for Chagas disease, using cryo and MD simulations. Unlike its *E. coli* counterpart (EcMscS), TcMscS has a minimal structure but retains mechanosensitive activity, causing membrane deformation at the protein-lipid interface. The study finds that ion conduction requires lipid displacement rather than major conformational changes in the channel, offering insights into lipid-mediated gating caused by TcMscS. The computational work is reasonably performed, with sufficient detail in the methods section, standard protocol usage, and basic analysis performed in replicate. The overall paper is clear to read. My suggestions on major revision relate to manuscript improvement include better integration of computational and wet lab data, as to further connect the experiments performed, utilize better subheadings to classify the work done, and consider a few additional analysis metrics to confirm the lack of conformational change resulting in the gating. Comments below:

Point 1: Improve the sub-heads to be completer and more directed at the outcomes of those sub sections.

Response: We agree with the reviewer and have now revised the subheadings to be more accurately directed at the outcomes of those sub sections. We have also rearranged the paragraphs of cryo-EM structure description (line 284-295), previously within the MD simulation section, to better separate and reflect each sub section. Below are the revised subheadings.

“Cryo-EM structure of TcMscS”

“Membrane deformation observed in MD simulations”

“Closed conformation mediated by pore lipids”

“Lipid blockade revealed by MD simulations”

Point 2: Improve integration of wet lab work to confirm simulation outcomes/movement of lipids.

Response: We thank the reviewer for this point. We have made efforts to improve integration of wet lab work to confirm simulation of outcomes/movements of lipids. For instance, we have the following descriptions in the main text.

Line 300:

“In the absence of lipids inside the pore, it remained fully hydrated, with a simulated unitary conductance of 393.6 ± 33.1 pS (Fig. 5a and Extended Data Fig. 5c), which is largely consistent with our measurements from excised membrane patches (478 ± 11 pS) and previous studies⁸.”

Line 303:

“In contrast, simulations with pore lipids showed seven POPC lipid tails within the pore, consistent with the observed lipid densities in the cryo-EM reconstructions, completely blocked

water and ion permeating, regardless of the presence or absence of an applied 500 mV voltage (Fig. 5b and Extended Data Table 4). These results further support that our TcMscS structures represent a lipid-occluded non-conducting state.”

Line 329:

“Inspired by these simulation results, we experimentally introduced the D47A mutation on the non-inactivating background C66L (D47A/C66L) and measured the activation pressure thresholds of C66L and D47A/C66L relative to that of the endogenous EcMscL channel. Consistently, the D47A/C66L mutant channel maintained the same unitary conductance (448.1 ± 11.5 pS, $N = 5$) as that of C66L, but was activated at a higher pressure threshold ($P_{MscL}/P_{D47A/C66L} = 1.95 \pm 0.51$, $N = 8$) than C66L ($P_{MscL}/P_{C66L} = 3.70 \pm 1.24$, $N = 8$).”

Point 3: Discuss limitations of study design related to MD simulations – force field selection impact, time scales, etc.

Response: We have now added the following paragraph in the ‘Conclusion’ section.

Line 431:

“Although partial delipidation in the pore was observed in multiple MD simulations under tension, a complete transition from the lipid-occluded nonconducting state to a conductive state was not captured within the short timescale of our simulations. In addition, the competition between lipids and water molecules inside the channel pore may be sensitive to the force field and thus requires more thorough investigation⁴⁹.”

Point 4: Perform analysis to confirm a lack of structural change more clearly – for example – you note that “Under 20 mN/m tension, two lipid tails exited the pore through the gaps between neighboring TM3a 14 helices around 0.5 μ s, but the pore was still blocked by the remaining lipid tails, without significant conformational changes of the channel.” How have you classified significant or not significant conformational change? PCA analysis could be beneficial here, as well as structural overlays of different time points to show lack of change (as well as comparison to initial cryo structure). If doing structural overlays, please perform clustering over time periods as to not cherry pick structures.

Response: We thank the reviewer for this suggestion. We have added PCA analysis and clustering analysis in the new Extended Data Fig. 5e. The overlaid protein backbones from each cluster centroid indicates minimum conformational changes except for the N helix.

Point 5: More discussion and demonstration of trends among the three replicates of data – in the SI each simulation cluster structure comparison to initial starting state should be shown so the reader can see this lack of conformational change in a structure perspective in addition to figure SI 5.

Response: We thank the reviewer for this point and have now included clustering analysis in the new Extended Data Fig. 5e, along with the alignment of the initial cryo-EM structure of TcMscS in lipid nanodisc.

Point 6: In the methods, it would be helpful to indicate how the channel protein was placed into

the membrane structure – what amino acids were used for anchoring at the leaflet interface to relate to experimental data for accurate placement. You should also include more details on how analysis was performed/packages/etc.,

Response: As shown in Extended Data Fig.5a, we have positioned the channel into multiple different locations within the membrane bilayer. We found that the final equilibrated channel-membrane position is independent of the starting positions. We have stated this observation in the main text.

Line 167:

“Simulations were initiated by placing the channel at different z-positions relative to the membrane center. In all three bilayers, the TcMscS channel converged within 100 ns to the same equilibrium position, wherein TM2 and TM3a are partially embedded in the bilayers and the N helices are anchored below (Fig. 1f and Extended Data Fig. 5a).”

To make it clear, we have also stated the following in Methods.

Line 572:

“Because the thickness of the presumed transmembrane domain of TcMscS is less than that of a typical lipid bilayer, we prepared multiple systems to ensure that channel–membrane interactions were not biased by the starting points. In these systems, the channel was either aligned using the Orientations of Proteins in Membranes (OPM) database⁶² or manually placed within the membrane at multiple different z-positions.”

Point 7: Minor – in the methods, you do not have molarity associated with your ion concentration.

Response: We thank the reviewer for this point and have now corrected this in the Methods section.

Line 586:

“The protein and bilayer systems are solvated by 150 mM KCl solution.”

Point 8: Data availability – please consider posting simulation input (config files, starting structure) and major (at least .pdb) output files onto Zenodo or other file sharing platforms.

Response: Input files, initial and final coordinates have been uploaded and are available at <https://github.com/LynaLuo-Lab/TcMscS>. We have now included the following in Data Availability.

Line 644:

“For molecular dynamics simulation data, initial coordinates, simulation input files, and a coordinate file of the final output are available at <https://github.com/LynaLuo-Lab/TcMscS>.”

Reviewer #2 (Remarks to the Author):

This manuscript reports exciting findings on TcMscS, a mechanosensing channel from *Trypanosoma cruzi* with significant therapeutic potential. The authors utilized a combination of structural, functional, and computational approaches to elucidate the structure, conformation, and gating mechanisms of TcMscS. Their results reveal unique structural features of TcMscS, dramatic membrane deformation induced by the channel, and evidence to support a lipid-mediated gating mechanism. These insights significantly advance our understanding of this important mechanosensing channel, provide novel insights into mechanosensing, and lay a foundation for the development of therapeutic strategies targeting *T. cruzi*. Overall, these findings are significant and novel, with high impact in the field.

Several comments that need to be addressed:

Point 1: The conformational state of the reported structure needs clarification. The author propose that the channel was captured in an open state. However, they also suggested ‘direct lipid-mediated mechanical gating transition without major conformational changes.’ If this is the case, would the ‘closed’ state differ significantly from the proposed open state? Potentially, the observed pore-blocking lipids may actually suggest a closed state. In addition, the cryo-EM samples in nanodisc were prepared without applied tension, which may potentially favor a closed conformation. In any case, please further clarify the conformational state of the current structure.

Response: We agree with the reviewer that the observed conformations represent a closed state because pore lipids were observed to block the ion conduction pathway. We had meant that, in the absence of pore lipids, the channel conformation appeared to be conductive. To make it clear, we have now changed the subheading from “*Open structure?*” to “*Closed conformation mediated by pore lipids*”. We have also further clarified in our main text that the TcMscS conformations represent a “lipid-occluded nonconducting state”.

Line 310:

“These results further support that our TcMscS structures represent a lipid-occluded non-conducting state.”

Point 2: Fig. 3a shows that TcMscS overlays well with the open conformation of EcMscS. Does this imply that TcMscS adopts a similar membrane-embedded position as the open conformation of EcMscS? Please comment on their membrane boundaries.

Response: We appreciate the reviewer for this point. We have now updated Fig. 3a with superposition of TcMscS and EcMscS with the intact, long TM1 helix to better indicate the membrane boundaries. Accompanying channel opening of EcMscS, the surrounding lipid membrane gets thinner. This does imply that TcMscS adopts a membrane-embedded position closer to that of the open EcMscS channel. We have now stated the following in our main text.

Line 231:

“suggesting that the TcMscS structure may represent an open conformation and that TcMscS adopts a membrane-embedded position closer to that of the open EcMscS channel.”

Point 3: In the coarse-grained simulation, it seems that the protein conformation was constrained during the simulations, while the lipids were not. Please clarify the rationale and comment on whether this might impact the lipids' relative movement in relation to the protein.

Response: This is a good point. Because of the protein backbone constraint in Martini coarse-grained model, we cannot rule out the possibility of this impacting the lipids' relative movement in relation to the protein. Therefore, we have completely removed our previous coarse-grained simulation data and focused on the partial delipidation observed in all-atom simulations.

Point 4: What is the effect of mutating the positive charged residues in the N-helix?

Response: We agree with the reviewer that the effect of mutating the positive charge residues in the N-helix is important to be investigated. We have examined three mutations (new Extended Data Fig. 7).

1. When the N-helix (N-terminal 16 amino acids) was deleted, the mutant channel was nonfunctional. We did not detect any channel activity from 28 patches.
2. For the K2A point mutation, we did not detect any channel activity from 16 patches.
3. For the R3A point mutation, the mutant channel remains mechanosensitive, with similar pressure threshold and unitary conductance to those of the wild type (new Extended Data Fig. 6d).

These results suggest that the N-helix plays an important role in channel function. The effect of the positive charge residues is complicated in that K2A is nonfunctional but R3A is similar to the wild type. For these reasons, we have decided to not elaborate the charge effect in the main text but have now included the results of these mutations in the extended data to be more transparent and complete (new Extended Data Figs. 6d, 7).

Point 5: The proposed competition between lipids and water is interesting. Under typical physiological conditions, is water always present, and how does this affect the competition?

Response: We thank the reviewer for this point. Under typical physiological conditions, both water and lipids are always present. We observed robust pore lipid densities in the cryoEM structures of channels isolated from the biological membranes by detergents. This indicates that, in the native membranes, lipid molecules already reside in the pore. Partial or complete delipidation of the pore is necessary for ion conduction. In our MD simulations, we show that the competition between pore lipids and pore water is sensitive to membrane tension and pore hydrophobicity (D47A). In addition, lipid types and force field may also influence such competition, which is interesting for further study. We have now also discussed the limitations of our simulation study in the Conclusion section.

Line 431:

“Although partial delipidation in the pore was observed in multiple MD simulations under tension, a complete transition from the lipid-occluded nonconducting state to a conductive state was not captured within the short timescale of our simulations. In addition, the competition between lipids and water molecules inside the channel pore may be sensitive to the force field and thus requires more thorough investigation⁴⁹.”

Point 6: Minor: In method section: ‘The purified TcMscS protein was concentrated to ~3.2 mg ml⁻¹ and mixed with soybean polar lipid extract and scaffold protein MSP1E1 at a molar ratio of 1:0.5:50 for 10 mins’. Please clarify the molar ratio among protein: lipid:MSP.

Response: We thank the reviewer for this point and have revised the manuscript.

Line 483:

“The purified TcMscS protein was concentrated to ~3.2 mg ml⁻¹ and mixed with scaffold protein MSP1E1 and soybean polar lipid extract at a molar ratio of 1:0.5:50, and incubated for 10 mins on ice.”

Reviewer #3 (Remarks to the Author):

Reviewer #4 (Remarks to the Author):

This study on lipid-mediated gating of a miniature mechanosensitive MscS channel reports cryo-EM structures of this channel, complemented with functional and computational investigations. The authors found that the unusually short transmembrane domain of this MscS channel results in a drastic deformation of the membrane around the channel. They observed in the cryo-EM maps lipid-like densities in the pore and that the occupancies of the modelled lipids decreases as a function of lateral tension and pore hydration in molecular dynamics simulations. Based on these observations, the authors propose a direct lipid-mediated mechanical gating transition that might happen without major conformational changes of the channel protein. This study is of interest to the broad readership of Nature Communications.

Fig 2:

The authors’ presented representative recordings but a more quantitative and statistically sound presentation is needed, i.e., quantification of observables with at least 3 biological replicates. In particular, the following would strengthen the authors’ argument.

Point 1: A pressure-response relation of TcMscS constructed using the authors’ setup.

Response: To be clear, we had conducted all our recordings with at least 3 biological replicates and the numbers of replicates were also indicated in the corresponding figures and figure legends (Extended Data Fig. 6d). We have now calculated the gating pressure threshold ratios for all constructs mentioned in the manuscript and updated the Extended Data Fig. 6d (Source Data File is also provided).

For the full pressure-response relationship, we could only measure using the non-inactivating background (C66L). Because of the continuous, fast, and irreversible inactivation of the WT

TcMscS channels, it is very difficult (if possible at all) to measure it for WT. A dose-response curve is obtained by running a relatively slow (to maintain quazi-equilibrium) pressure ramp from zero to the point of current saturation (i.e. activation of all channels in the patch, when further increase of tension does not elicit extra current). This curve is then fitted by Boltzmann function, etc. Unfortunately, there are a number of obstacles that prevent us from doing this (otherwise routine for channels like EcMscS) analysis:

1) WT TcMscS inactivates quickly and irreversibly. We have to discard patches that take more than a minute to form at a very mild suction (-10 mmHg or less), otherwise we do not observe any TcMscS activity. For EcMscS however, it is not uncommon to apply much higher negative pressures (up to -100 mmHg) for much longer times (~10 min) to get a high-resistance membrane seal without affecting MscS properties.

2) Therefore, typically in our excised patches, only a few (<5) channels were present. This had prevented us to measure macroscopic currents and fit them with a Boltzmann function - which would need to be done for statistical analysis.

3) After patch formation and patch excision, the inactivation process of TcMscS continues (as shown in new Extended Data Fig. 6a), which takes a couple of minutes and makes application of slow pressure ramps essentially senseless.

4) One could however apply short (~2 s) pressure ramps, taking into account all the reservations concerning quazi-equilibrium states. However, we cannot do it manually (with reproducible rate and quality) in our experimental setup.

Point 2: Statistical analysis of the 'inactivation' in the absence of sustained tension.

Response: If we understand correctly, the reviewer requests statistical analysis of the inactivation rates in the absence of sustained tension. This is an important point but we could not conduct these experiments. To measure the inactivation rates, very short pulses of transmembrane pressure need to be applied over time to test how many channels can be gated after certain time upon patch excision. Not all the channels in the patch would be activated at the same time. In addition, saturated currents (i.e. all the channels present in the patch are activated at the same time) are difficult to achieve when the channels are continuously (and rather quickly and irreversibly) inactivating. Also, the inactivation process may be tension dependent. These difficulties have limited our capability to evaluate the detailed 'inactivation' process, which we believe is beyond the scope of our current work.

Point 3: Statistical analysis of the inactivation during sustained tension of TcMscS and TcMscS-C66L.

Response: We have the same challenges as elaborated above in **Response to Point 2**.

Point 4: Line 220-222, 'Moreover, the identical structures...inactivated conformation.': The observation that a seemingly activating mutant does not change the conformation alone provides little support for suggesting or rejecting whether the determined structure is or is not in an inactivated conformation.

Response: To clarify, C66L is a non-inactivating mutant with higher tension sensitivity, therefore the observed conformation is unlikely in an inactivated state. We have now also weakened our claim and stated the following.

Line 223:

“Moreover, the identical structures of the wild-type channel and a non-inactivating mutant, C66L, indicate that our cryo-EM structures unlikely represent an inactivated conformation.”

Point 5: Line 268-270, ‘In contrast, introduction of a bulkier hydrophobic residue...’:
No data have been provided in support of this statement.

Response: The L97F mutation, with a bulkier hydrophobic residue introduced, rendered a non-conductive channel. We attempted to measure L97F activity but observed no mechanosensitive currents. We have now also included these negative data (electrophysiological recordings) in the Extended Data Fig. 7. Tension-activated currents were not observed for L97F, and only MscL channels, used as an internal control, were activated by near-lytic tension in these excised membrane patches.

Point 6: Line 270-273, ‘These results indicate that L97 defines...’:
The authors have not provided any data showing that L97 does not modify the pressure-response relation – an effect of L97 on mechanical gating would be manifested in a change in the conformational equilibrium under tension.

Response: We had stated that these mutants maintain the same gating pressure threshold as that of the wild type. We have now also plotted the gating pressure thresholds for all three L97 mutations, including L97G, L97A, and L97V, in the new Extended Data Fig. 6d.

Point 7: Line 282-285, ‘Estimation of the pore radius...’ and Fig 4b:
The lipid lined pore radius is almost 3 Angstroms and seems to correspond to where polar lipid headgroups are situated, which might be sufficient for the passage of hydrated physiological ions. Perhaps the authors can provide more description/explanation on the lipid gating mechanism.

Response: Our simulations show that the pore remains non-conductive because lipid tails are flexible and dynamic within the pore, which blocks water and ion permeation, despite the 3 angstrom radius. We have now summarized our data in Extended Data Table 4.

Point 8: Fig 4c:
The authors could overlay the pore radius of TcMscS without modelled lipids to show the difference that lipid occupancy introduces.

Response: We have shown the TcMscS pore profiles without modeled lipids in Fig. 3d, and with modeled lipids in Fig. 4c. We think that an additional overlay of these profiles seems redundant.

Point 9: Line 316, ‘These simulations suggest that the pore is conductive in the absence of pore lipids.’:

Related to my comment above on lipid gating, perhaps the authors could perform the same voltage simulations to estimate the conduction properties of the lipid-lined state with the bound lipids restrained, which would provide further support to the lipid gating model.

Response: Per the reviewer’s suggestion, we performed voltage simulations with tension and found that the pore remains occluded by lipid tails. This is not surprising as the position of neutral lipid tails in the pore is not expected to be changed by voltage. We have summarized our results in the new Extended Data Table 4.

Point 10: Line 323-328, 'To corroborate the simulation results, we...':
Please provide the associated experimental data with at least 3 biological replicates.

Response: We had conducted all our recordings with at least 3 biological replicates. We have now included the gating pressure threshold ratio and unitary conductance for all constructs mentioned in the manuscript and updated the Extended Data Fig. 6d, including mutants D47A/C66L and C66L. We have also stated the following in the main text.

Line 409:

“Consistently, the D47A/C66L mutant channel maintained the same unitary conductance (448.1 ± 11.5 pS, $N = 5$) as that of C66L, but was activated at a higher pressure threshold ($P_{MscL}/P_{D47A/C66L} = 1.95 \pm 0.51$, $N = 8$) than C66L ($P_{MscL}/P_{C66L} = 3.70 \pm 1.24$, $N = 8$).”

Reviewer #5 (Remarks to the Author):

(Remarks to the Authors):

Using cryo-electron microscopy, electrophysiology and molecular dynamics simulations, Zhang and colleagues found that TcMscS (in detergent and nanodiscs) has two short membrane-embedded helices that do not fully span the lipid bilayer and a unique short N-terminal domain as well as a different pore architecture (ion permeation pathway), causing significant membrane deformation. This results in a funnel-shaped bilayer around the channel, where resident lipids block the ion permeation pathway. Ion conduction requires these lipids to be displaced by lateral membrane tension. Further, patch-clamp measurements and mutagenesis studies suggest that TcMscS gating is directly mediated by lipids, without major conformational changes in the protein. These findings provide insight into mechanosensitive channel function.

The proposed data are clearly novel, and the structure(s), for the first time, highlight unique structural features of the minimal orthologue of MscS channels, as detailed above. These findings are of great significance, given the involvement of this channel in the biological process of protozoan development and infectivity due to its mechanosensitive features. Importantly, the structural and functional data presented in this manuscript provides a structural framework for the development of specific drugs to target/inhibit TcMscS channels.

In my opinion, the methodology used, as the well as data presentation and quality of the data seem very high, with all necessary (supporting) files provided to judge the quality of the data. Although I am not an expert on cryo-electron microscopy, the provided density maps and quality reports seem convincing. Overall, the manuscript is very well written with clear figures and is of great interest to the readership of Nat. commun.

In light of the unique structural features of TcMscS and its biological importance, I would be happy to support the acceptance of this manuscript. However, some major concern (discussed below) related to the functional and MD approaches needs to be addressed.

Point 1: The significance of the N-terminus has only been investigated through in-silico experiments (MD simulations). However, it is speculated that the positive charges attract the negative charges of the lipid head groups of the inner membrane leaflet, which in turn has a

strong impact on the deformation of the bilayer. However, removing the entire N-terminal region could induce various conformational changes, which in turn could lead to a volume change and consequently deformation. Whether the effect is truly due to specific charge interactions could best be demonstrated by neutralizing the charges and measuring the response to mechanical stimuli. In my view, these electrophysiological measurements would strengthen the manuscript.

Response: We agree with the reviewer that neutralizing the charges in the N helix and measuring the response to mechanical stimuli is important to strengthen the manuscript. We have examined three mutations.

1. When the N-helix (N-terminal 16 amino acids) was deleted, the mutant channel was nonfunctional. We did not detect any channel activity from 28 patches.
2. For the K2A point mutation, we did not detect any channel activity from 16 patches.
3. For the R3A point mutation, the mutant channel remains mechanosensitive, with similar pressure threshold and unitary conductance to those of the wild type (new Extended Data Fig. 6d).

These results suggest that the N-helix plays an important role in channel function. The effect of the positive charge residues is complicated in that K2A is nonfunctional but R3A is similar to the wild type. For these reasons, we have decided to not elaborate the charge effect in the main text but have now included the results of these mutations in the manuscript to be more transparent and complete (new Extended Data Figs. 6d, 7).

Point 2: Another critical point is that TcMscS is resolved in the open state (in comparison to EcMscS), with the permeation pathway seemingly blocked by pore lipids, which exit to the extracellular funnel-shaped space due to lateral membrane tension. A direct demonstration that this clears the channel pore and allows K⁺ permeation would be to block the visible current, potentially also in the non-inactivating C66L mutant. Additionally, how can the observed desensitization to mechanical stimuli be explained in the context of this lipid-induced pore plug mechanism?

Response: TcMscS, like other MscS homologs, is not a K⁺-selective channel but rather a non-selective channel with a slight preference for anions. Moreover, there are no MscS-specific pore blockers. The toxin GsMTx4, which is often used to inhibit MscS and a few other mechanosensitive channels, seems to act at the protein-lipid interface and modify mechanical gating. Gadolinium ions inhibit MscS channels by altering membrane properties, rather than a “plug”, physically blocking the ion channel pore.

References:

1. Gnanasambandam R et al, GsMTx4: Mechanism of Inhibiting Mechanosensitive Ion Channels. *Biophys J* 112:31-45 (2017). PMID: 28076814; PMCID: PMC5231890.
2. Ermakov YA et al, Gadolinium ions block mechanosensitive channels by altering the packing and lateral pressure of anionic lipids. *Biophys J* 98:1018-27 (2010). PMID: 20303859; PMCID: PMC2849073.

In our studies, we measured TcMscS currents using *E. coli* giant spheroplasts prepared from the well-established MJF516 cell line, in which endogenous mechanosensitive channels, including MscK, MscS, YbiO, and YjeP, were deleted. Therefore, the TcMscS (and its mutants) currents are readily to be identified by its characteristic unitary conductance.

For tension-induced desensitization, our hypothesis is that the channel protein, upon activation, adopts a conductive conformation as observed in our cryo-EM structures but membrane lipids may block the channel pore via distinct pathways and induce a desensitized or inactivated state. Alternatively, upon channel activation, the channel protein transits into a non-conductive conformation (similar to MscS) that allows lipid molecules to irreversibly block the conduction pathway. In both cases, channel opening and irreversible closure occur via distinct conformational trajectories. Because we do not have structural data supporting these hypotheses, we would like to stay away from stating these possibilities. These could be potentially resolved by future studies of functional mutants of TcMscS and/or other parasite MscS orthologs.

Point 3: Another point concerns the competition between pore lipids and water within the channel pore. While the difference from E. coli MscS channels has been described, indicating that the TcMscS pore is less hydrophobic but still hydrophobic enough to exclude water, it remains unclear how this competition could arise. Does the description of water refer to the pore from the intracellular side up to F64 or water in the extracellular pore?

Response: We thank the reviewer for this point. We observed robust pore lipid densities in the cryoEM structures of channels isolated from the biological membranes by detergents. This indicates that, in the native membranes, lipid molecules already reside in the pore. Partial or complete delipidation of the pore, subsequently allowing water entering the pore, is necessary for ion conduction. In our MD simulations, we show that the competition between pore lipids and pore water (in the extracellular pore) is sensitive to membrane tension and pore hydrophobicity (D47A). The intracellular portion of the pore up to F64 is already filled with water molecules. Our updated new Fig. 5a and Fig. 5b better illustrate these points.

Point 4:

4a: The figures could benefit from additional information and details. While most details are provided in the legends, some are only mentioned in the methods section, making the visual representation not always self-explanatory. For example, in Figure 2, it is unclear what currents are being measured, at what potential, and which ion species are involved.

Response: All the experimental details have been provided in the Methods, and we have now also included more details in the figure legends to make visual representation more self-explanatory. In Figure 2, like in all other patch-clamp recordings, symmetric KCl buffer (200 mM KCl, 90 mM MgCl₂, 2 mM CaCl₂, and 5 mM HEPES pH 7.2) was used, and all measurements were carried out at -30 mV membrane potential. Because TcMscS, like other MscS-like channels, is non-selective with a slight preference for anions, the conductive ions include K⁺, Cl⁻, Ca²⁺, and Mg²⁺. We have now included the following in the figure legends.

“Symmetric KCl buffer (200 mM KCl, 90 mM MgCl₂, 2 mM CaCl₂, and 5 mM HEPES pH 7.2) was used, and all measurements were carried out at -30 mV membrane potential.”

4b: Additionally, some details appear contradictory. Figure 2a does not clearly indicate that TcMscS transitions into an inactivated state upon repetitive pulses. Instead, a high negative pressure (-120 mmHg) seems to result in reduced activation. A recording showing repetitive pulses at the same pressure, where channel activity decreases over time, appears to be missing.

Response: For stretch-activated channels such as TcMscS, application of higher membrane tension, as measured by increased pressure in the recording pipette, under quasi-equilibrium

conditions, should elicit higher amplitudes of currents in the absence of activation-dependent inactivation. Therefore, the observed reduced currents with repeated pressure ramps with increased pressure indicate channel inactivation or desensitization. We used relatively slow pressure ramps with the same pressure application speed (about 4 mmHg per second, “quasi-equilibrium conditions”) to reliably identify the first opening of TcMscS channels for estimation of their tension sensitivity. Under these conditions, the increase of the pressure at the same rate would induce activation of more channels until all of them are active at saturating pressure. However, as it was shown in Figure 2a, this was not the case. Higher tension (or pressure) did not result in activation of more channels; in fact the number of open channels decreased. Ultimately, even near-lytic tension (very high membrane tension), in which the endogenous MscL channels were activated, no more TcMscS currents were observed. These observations clearly indicated irreversible channel inactivation of TcMscS and ultimately even the highest possible membrane tensions failed to activate TcMscS channels.

We also have recordings with repetitive pressure ramps of the same amplitude (now provided in new Extended Data Fig. 6a).

4c: Furthermore, the endogenous reference channel is described as having a high activation threshold compared to TcMscS. However, in Figure 2b, it appears to be activated at -95 mmHg, the same pressure that produces robust TcMscS activation in Figure 2a. How can mixed activity be ruled out in this case? Was a calibration performed, for example, by heterologously expressing a non-functional TcMscS channel (dominant-negative mutant) in the cell line and determining the pressure profile of the endogenous EcMscS?

Response: To clarify, these mechanosensitive channels are activated by lateral membrane tension (not directly by the transmembrane pressure). Tension γ is proportional to both the transmembrane pressure (ΔP , mmHg) and the radius of the spherical membrane (r).

$$\gamma = \Delta P r/2$$

Each patch (even formed in similarly sized patch-pipettes) has a different radius and therefore the same transmembrane pressure applied to different patches induces different lateral membrane tensions. This is exactly the reason to include the endogenous reference channel MscL with known tension activation as an internal calibration. MscL has a very high unitary conductance (~ 3.3 nS in the experimental buffer, described in Methods) and is activated by high near-lytic tension. Moreover, its activation is generally insensitive to environmental factors including ionic conditions and pH etc. Figures 2a and 2b illustrate two different patches with different radii, where MscL is activated by different transmembrane pressure but in principle the same lateral membrane tension. Therefore, to characterize tension sensitivity, we expressed TcMscS and its mutants in a well-characterized cell line, MJF516, tailored for studies of mechanosensitive channels. This cell line lacks the biggest and the most expressed mechanosensitive channels MscS, MscK, YbiO, and YjeP, but still contains endogenous MscL, which was used as an internal tension calibration. Tension sensitivity of the WT TcMscS and mutants was evaluated using the ratio of $P_{\text{MscL}}/P_{\text{TcMscS}}$. It was calculated as the ratio of pressures at which the first MscL and TcMscS channels were activated, respectively, in response to a slow ~ 4 mmHg s^{-1} pressure ramp.

Reviewer #6 (Remarks to the Author):

In the manuscript “Lipid-mediated gating of a miniature mechanosensitive MscS channel from *Trypanosoma cruzi*” by Zhang et al the authors collected structural data for TcMscS, used molecular dynamic simulations to uncover the channel gating dynamics. This manuscript gives a deep insight into the gating structure of TcMscS through new structures, new models that observe lipid movement during channel gating, and patch clamp electrophysiology where single channels were observed. In general I found the manuscript to be well written and of interest to the broad scientific community as well as the mechanosensitive ion channel community. The methods section was clear and would allow for other scientists to complete similar work. I do have several major and minor corrections as outlined below.

Major

Point 1: Patch clamp electrophysiology of point mutations and chimeras:

-This data is interesting but incomplete. In some cases the pressure threshold is reported for some mutations but not all mutations (lines 326-328). This value is critical in the comparison of point mutations to the wild type. Additionally, you report the unitary conductance for some mutations but not all can you please include this value for all point mutations, the wild type, and any chimeras that you discuss in the manuscript.

Response: We acknowledge this point and have now included gating pressure threshold and unitary conductance for all mutations mentioned in the manuscript (new Extended Data Fig. 6d) as well as a source data file in spreadsheet.

Point 2: For each mutation that you created and discuss in the manuscript can you show a representative trace?

Response: We have now included representative traces for all the mutations (new Extended Data Fig. 7), including L97G, L97A, L97V, L97F (no activity, MscL currents only), R3A, dN16, K2A (no activity, MscLs only), and D49A-C66L. Representative traces for the rest of the constructs (WT, C66L, TcMscS-Ec, TcMscS-Vc) have already been provided in the original manuscript.

Point 3: In line 269 you report that L97F does not conduct in response to applied tension, this would implies that the channel is surface expressed. Can you report how you confirmed that this mutant was surface expressed? Additionally, can you show a representative trace for the data collected?

Response: We thank the reviewer for this point. We have now included a representative trace of the data collected from the spheroplasts expressing L97F. Although we did not independently confirm the presence of L97F in the spheroplast’s membrane, we assume its expression is similar to all the other L97 mutations including L97G, L97A, and L97V. We originally expressed TcMscS channel with a C-terminal EGFP tag to confirm surface expression. However, we had to remove the C-terminal EGFP tag to observe tension elicited currents. Therefore, we cannot verify surface expression using fluorescence. Alternatively, we could isolate the inner *E. coli* membrane fraction and verify surface expression by Western Blots using anti-TcMscS antibodies, which are not available. Adding other tags for Western may result in non-functional channels like the C-terminal EGFP fusion constructs. Therefore, verification of surface expression is not trivial. We agree that these results, if feasible, will strengthen, but not significantly change our conclusions.

Point 4: in Figure 2 panel b and c it appears that the channels display continued activity after

the tension is removed, does this suggest that the channel has difficulty closing if the tension is removed prior to the channel entering the desensitized state?

Response: The reviewer is correct that TcMscS displays continued activity after the tension is removed. It is common that MscS-like channels display continued activity after removal of tension. For instance, such channel behavior has been reported for EcMscS, MscCG, and MSL10.

References:

1. Ridone P, Nakayama Y, Martinac B, Battle AR. Patch clamp characterization of the effect of cardiolipin on MscS of *E. coli*. *Eur Biophys J*. 2015 Oct;44(7):567-76. doi: 10.1007/s00249-015-1020-2. Epub 2015 Apr 5. PMID: 25842033.
2. Nakayama Y, Yoshimura K, Iida H. Electrophysiological characterization of the mechanosensitive channel MscCG in *Corynebacterium glutamicum*. *Biophys J*. 2013 Sep 17;105(6):1366-75. doi: 10.1016/j.bpj.2013.06.054. PMID: 24047987; PMCID: PMC3785867.
3. Maksaev G, Haswell ES. MscS-Like10 is a stretch-activated ion channel from *Arabidopsis thaliana* with a preference for anions. *Proc Natl Acad Sci U S A*. 2012 Nov 13;109(46):19015-20. doi: 10.1073/pnas.1213931109. Epub 2012 Oct 29. PMID: 23112188; PMCID: PMC3503204.

Point 5: Lines 326-328: You reference the pressure threshold for two mutants without inclusion of the WT channel threshold, this makes it difficult to compare the change in threshold to the WT channel. Can you please include the pressure threshold for this channel and representative traces for these mutations?

Response: This is a good point. We have now reported gating pressure thresholds for all the constructs mentioned in our manuscript (new Extended Data Figure 6d). C66L has a significantly lower gating threshold than the wild type. The tension sensitivity of D47A-C66L is similar to that of the wild type. The unitary conductance of D47A-C66L is between those of the wild type and C66L.

Point 6: Extended Data Figure 6: Please include unitary conductance values for all mutations studied in the manuscript.

Response: We accept this point and have now included the unitary conductance for all the constructs mentioned in the manuscript (new Extended Data Fig. 6d) except TcMscS-Ec as it is very noisy and does not allow reliable identification of individual channel openings (as illustrated in Extended Data Fig. 6b).

Minor

Point 7: Line 473: update the text to show how threshold values were calculated for all mutants

Response: We have now updated the methods to show how threshold values were calculated for all mutants.

Line 567:

“The measurements were averaged from multiple (at least three) patches expressing the WT TcMscS or mutations.”